# Pupil size reflects activation of subcortical ascending arousal system nuclei during rest

**Beth Lloyd[1]\*, Lycia D de Voogd[2,3], Verónica Mäki-Marttunen[1], Sander Nieuwenhuis[1]**

[1]Institute of Psychology, Leiden University, Leiden, Netherlands; [2]Donders Institute, Centre for Cognitive Neuroimaging, Radboud University Nijmegen, Nijmegen, Netherlands; [3]Behavioural Science Institute, Radboud University, Nijmegen, Netherlands

**Abstract** Neuromodulatory nuclei that are part of the ascending arousal system (AAS) play a crucial role in regulating cortical state and optimizing task performance. Pupil diameter, under constant luminance conditions, is increasingly used as an index of activity of these AAS nuclei. Indeed, task-based functional imaging studies in humans have begun to provide evidence of stimulus-driven pupil-AAS coupling. However, whether there is such a tight pupil-AAS coupling during rest is not clear. To address this question, we examined simultaneously acquired resting-state fMRI and pupil-size data from 74 participants, focusing on six AAS nuclei: the locus coeruleus, ventral tegmental area, substantia nigra, dorsal and median raphe nuclei, and cholinergic basal forebrain. Activation in all six AAS nuclei was optimally correlated with pupil size at 0–2 s lags, suggesting that spontaneous pupil changes were almost immediately followed by corresponding BOLD-signal changes in the AAS. These results suggest that spontaneous changes in pupil size that occur during states of rest can be used as a noninvasive general index of activity in AAS nuclei. Importantly, the nature of pupil-AAS coupling during rest appears to be vastly different from the relatively slow canonical hemodynamic response function that has been used to characterize task-related pupil-AAS coupling.

**\*For correspondence:**
b.lloyd@fsw.leidenuniv.nl

**Competing interest:** The authors declare that no competing interests exist.

## Editor's evaluation

These are important findings that show that pupil size is not only governed by the locus coeruleus but also by other neuromodulatory subcortical systems. Furthermore, the authors demonstrate that using a standard hemodynamic response kernel is not appropriate for capturing the activity of these systems, at least at rest. Thus, this paper presents compelling evidence against two prevalent working assumptions among researchers in the field.

## Introduction

Neuromodulatory brainstem, midbrain, and basal forebrain nuclei that together form the core of the ascending arousal system (AAS) are situated deep in the brain. They have widespread projections to the cortex, making them ideally suited to alter cortical states and optimize task performance (*Bunzeck and Düzel, 2006*; *de Gee et al., 2017*; *Shine et al., 2021*; *Thiele and Bellgrove, 2018*). Pupil diameter, under constant luminance conditions, has vastly been used as a proxy for activity of these subcortical nuclei (*Joshi and Gold, 2020*). Indeed, task-related activity of these nuclei is accompanied by changes in pupil size and its first-order derivative (i.e., rate of change), as evidenced by

animal studies and functional magnetic resonance imaging (fMRI) studies in humans (*Cazettes et al., 2021*; *de Gee et al., 2017*; *Murphy et al., 2014*; *Varazzani et al., 2015*; *Yang et al., 2021*). Animal studies have also found pupil-AAS coupling of spontaneous fluctuations during rest (*Joshi et al., 2016*; *Reimer et al., 2016*). However, it is still largely unclear whether similar coupling can be found between resting-state fluctuations of pupil size and blood oxygen level-dependent (BOLD) signals in the human AAS. Assessing if and how activity in neuromodulatory brainstem, midbrain, and basal forebrain nuclei can be inferred from pupil size measurements is relevant for promoting our scientific and clinical understanding of AAS function.

A small number of human resting-state fMRI studies have investigated the brain activity associated with fluctuations in pupil size (*Breeden et al., 2017*; *Mäki-Marttunen and Espeseth, 2021*; *Murphy et al., 2014*; *Yellin et al., 2015*) and pupil derivative (*DiNuzzo et al., 2019*; *Schneider et al., 2016*). Their results with respect to a coupling between pupil size and AAS activity are inconclusive, with most studies not reporting evidence for such a relationship. However, the majority of these studies did not focus on the AAS. Moreover, they did not include specific localization methods to delineate AAS regions of interest (ROIs) or correct for physiological sources of noise, such as cardiac and respiratory fluctuations. These approaches are important for reliable measurements in these subcortical nuclei (*Brooks et al., 2013*; *Matt et al., 2019*). To date, only *Murphy et al., 2014* specifically investigated the relationship between pupil size and one AAS nucleus, namely the locus coeruleus (LC). The authors found a positive coupling between fluctuations in pupil size and activation in the LC during rest. To our knowledge, there have been no human fMRI studies so far that have reported a relationship between pupil size and other AAS nuclei during rest, despite the growing evidence from animal studies (*Joshi et al., 2016*; *Reimer et al., 2016*) speaking for such a relationship. Therefore, in this study, we aimed to investigate whether pupil size (and the pupil derivative) can be used as an index of activity in neuromodulatory AAS nuclei during rest.

To address this aim, we systematically examined simultaneous measurements of resting-state fMRI and pupil size from a large sample of healthy adults (N = 74). We monitored BOLD signal from a number of subcortical nuclei part of the AAS and implicated in the control of cortical arousal levels: the LC, the ventral tegmental area (VTA), dopaminergic substantia nigra (SN), the dorsal (DR) and median (MR) raphe nuclei, and the nucleus basalis of Meynert in the cholinergic basal forebrain (BF). Due to their size and location in the brain, studying these small nuclei using fMRI comes with a unique set of challenges (*Forstmann et al., 2016*; *Liu et al., 2017*; *Matt et al., 2019*). Here, we mitigated these challenges by implementing a number of methods, including multi-echo imaging to increase signal-to-noise ratio in subcortical structures (*Miletić et al., 2020*; *Puckett et al., 2018*; *Turker et al., 2021*), neuromelanin-weighted T1 imaging for delineation of the LC (*Clewett et al., 2016*; *Keren et al., 2015*; *Mäki-Marttunen and Espeseth, 2021*; *Priovoulos et al., 2018*), optimized brainstem co-registration (ANTs SyN; *Ewert et al., 2019*), physiological noise regression to suppress respiratory and cardiac artifacts (*Glover et al., 2000*; *Harvey et al., 2008*), and no spatial smoothing of fMRI data (*de Gee et al., 2017*).

As a first analysis, we intended to reproduce the analyses from two previous studies that did (*Murphy et al., 2014*) and did not (*Schneider et al., 2016*) find AAS correlates of pupil size during rest. The analyses in these studies were performed under the assumption that the relationship between pupil size and resting-state BOLD activity in AAS nuclei is governed by the canonical hemodynamic response function (HRF). As we could not replicate previous findings, we reasoned it is possible that during rest, when there is no external stimulus driving neural activity, the temporal relation between pupil dilation and AAS activity does not follow an HRF-like waveform. Therefore, we began examining the temporal relationship between pupil time series and AAS-BOLD activation using various transfer functions based on the canonical HRF, taking into account that subcortical structures have been characterized by faster time-to-peak (TTP) of the HRF than the cortex (*Lau et al., 2011*; *Lewis et al., 2018*; *Yen et al., 2011*). It is possible that the HRF does not provide an adequate model of the relationship between resting-state fluctuations in pupil size and AAS BOLD activity. We therefore also explored cross-correlations between AAS BOLD activity and the unconvolved pupil time series, systematically varying the forward and backward lag between the two measures. Together, these analyses offer new insights into the use of pupil size as an index of activity in AAS regions.

# Results

This section is organized as follows. We first report a couple of verification analyses aimed to ensure that we could replicate the resting-state correlations between pupil size and whole-brain BOLD patterns reported in previous studies and assess the signal quality within the subcortical nuclei. Then, we attempt to reproduce the pupil-LC coupling that was reported in *Murphy et al., 2014* by applying their convolution approach and LC localization method, as well as by interrogating the signal within our group LC ROI. After this, we move on to report three key analyses of pupil-AAS coupling aimed at understanding the temporal relationship between the two as well as the nature of this relationship: (i) an analysis in which we account for region- and participant-specific HRF differences in the convolution approach of the pupil time series; (ii) an analysis in which we explore pupil-AAS coupling while systematically adjusting the TTP of the HRF; and (iii) a cross-correlation analysis and cross-spectral power density analysis in which we explore the possibility that, during rest, the temporal relationship between pupil size and AAS BOLD patterns is not mediated by the HRF typically used in event-related fMRI design.

## Whole-brain pupil-BOLD patterns consistent with previous studies

We aimed to verify that our data showed the expected pupil-associated BOLD response patterns at the level of the cortex, cerebellum, and subcortical parts of the limbic system. To this end, we followed as closely as possible the approaches from two previous studies reporting pupil-BOLD coupling during rest (*Murphy et al., 2014*; *Schneider et al., 2016*) and indeed largely replicated their findings. First, following the approach by *Schneider et al., 2016*, we used pupil size (1 s shift) as a regressor in a GLM and convolved it with the canonical HRF (i.e., 6 s TTP). The assumption here is that the neural activity associated with spontaneous changes in pupil size is transformed into resting-state BOLD signal according to the same impulse response function as that driving neurovascular coupling during task performance. Indeed, we found positive correlations in the thalamus and negative correlations in the visual cortex and sensorimotor areas, as well as in the precuneus, cuneus, insula, superior temporal gyrus, and parahippocampal gyrus. These patterns of activation are consistent with those reported by *Schneider et al., 2016* (*Figure 1A* and *Table 1*). Second, we carried out the analysis in line with *Murphy et al., 2014*, using pupil size (not shifted) as a regressor

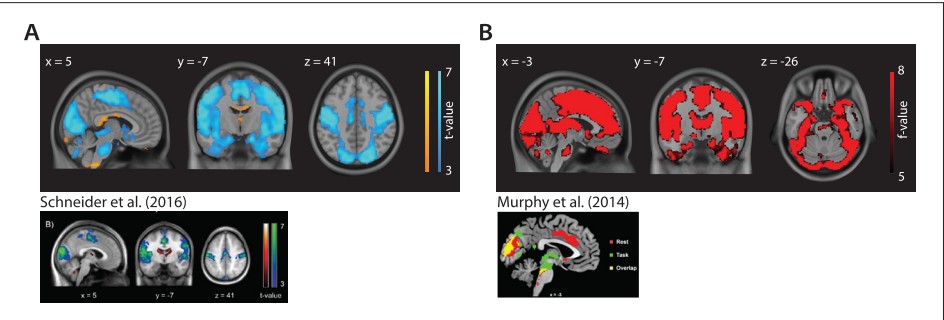

**Figure 1.** Whole-brain pupil-blood oxygen level-dependent (pupil-BOLD) coupling in comparison to previous studies. Neural correlates of pupil size from the analysis using the convolution approach (**A**) reproduced from Figure 1B of *Schneider et al., 2016* and (**B**) reproduced from Figure 2A of *Murphy et al., 2014*. Note that we only refer to the red and yellow activation in the figure from *Murphy et al., 2014*. Statistical parametric maps are thresholded at p<0.001, uncorrected, for visualization purposes only. Whole-brain cluster-level family-wise error (FWE)-corrected inferential statistics, in MNI space, are reported in *Tables 1 and 2* (n = 70). Statistical maps used to generate this figure is available in *Figure 1—source data 1*.

The online version of this article includes the following source data for figure 1:

**Source data 1.** Statistical maps used in *Figure 1*, *Tables 1 and 2*.

**Table 1.** Regions showing pupil-BOLD coupling using the convolution methods from *Schneider et al., 2016*.

| | Cluster | |
|---|---|---|
| | K | p$_{corr}$ |
| *Positive* | | |
| Thalamus (R), posterior cingulate cortex (R/L) | 871 | <0.001 |
| Rectus (L) | 199 | 0.010 |
| Cerebellum (R) | 142 | 0.043 |
| Cerebellum crus (L) | 250 | 0.003 |
| *Negative* | | |
| Cerebellum crus (R/L), cerebellum 6 (R/L), cerebellum 4/5 (R/L), lingual gyrus (R/L), calcarine (R/L), fusiform gyrus (R/L), cuneus (R/L), precuneus (R/L), cerebellum 4+5 (L), cerebellar vermis, hippocampus (R/L), parahippocampal gyrus (R), amygdala (R/L), thalamus (R), superior occipital gyrus (R/L), middle occipital gyrus (R/L), inferior occipital gyrus (R/L), superior parietal gyrus (L), inferior temporal gyrus (R/L), middle temporal gyrus (R/L), superior temporal gyrus (R/L), insula (R), postcentral gyrus (L), precentral gyrus (L), paracentral lobule (R/L), supplementary motor area (R/L), middle cingulate gyrus | 65223 | <0.001 |

Reported clusters survived whole-brain family-wise error (FWE) correction at the cluster level (p$_{FWE}$=0.05). Source data for *Table 1* available in *Figure 1—source data 1*.

R, right; L, left; p$_{corr}$, whole-brain-corrected cluster p-values; k, cluster size; BOLD, blood oxygen level-dependent.

and convolving it with the canonical HRF (i.e., 6 s TTP) as well as its temporal and dispersion derivatives (*Figure 1B* and *Table 2*). Adding these derivatives allows the timing of the HRF response peak and the width of the HRF response to vary across the whole brain. Here, we found significant clusters in the visual cortex, the insula, the anterior cingulate gyrus, and the inferior frontal gyrus, consistent with what was reported by *Murphy et al., 2014*. Overall, we found pupil-related BOLD response patterns across the whole brain that were highly consistent with the ones reported by the previous two studies.

Following *Murphy et al., 2014* approach, we also inspected pupil-associated activity in the LC. However, we did not find significant voxels when using our group LC mask as an ROI or when we applied the more liberal mask (*Keren et al., 2009*) used by *Murphy et al., 2014*. Thus, contrary to *Murphy et al., 2014*, we were unable to replicate pupil-LC BOLD coupling using the same convolution methods.

**Table 2.** Regions showing pupil-BOLD coupling using convolution methods from *Murphy et al., 2014*.

| | Cluster | |
|---|---|---|
| | K | p$_{corr}$ |
| Middle occipital gyrus (R/L), superior occipital gyrus (R/L), calcarine gyrus (R/L), cuneus (R/L), precuneus (R/L), angular gyrus (R/L), fusiform gyrus (R/L), cerebellum (R/L), middle temporal pole (R/L), inferior temporal pole (L), insula (R), inferior parietal lobule (R/L), superior parietal lobule (L), postcentral gyrus (R/L), middle frontal gyrus (R/L), medial frontal gyrus (R/L), inferior frontal gyrus (R/L), superior frontal gyrus (R/L), posterior cingulate gyrus, middle cingulate gyrus, anterior cingulate gyrus, supplementary motor area (R/L), middle frontal orbital (R/L), inferior frontal orbital (R/L), cerebellum crus II (R/L), cerebellum crus I (R/L), cerebellum 8 (R/L), cerebellum 9 (R/L), Pons | 91471 | <0.001 |
| Rectus (R/L) | 146 | 0.010 |

Reported clusters survived whole-brain family-wise error (FWE) correction at the cluster level (p$_{FWE}$=0.05). Source data for *Table 2* is available in *Figure 1—source data 1*.

R, right; L, left; p$_{corr}$, whole-brain-corrected cluster p-values; k, cluster size;.BOLD, blood oxygen level-dependent.

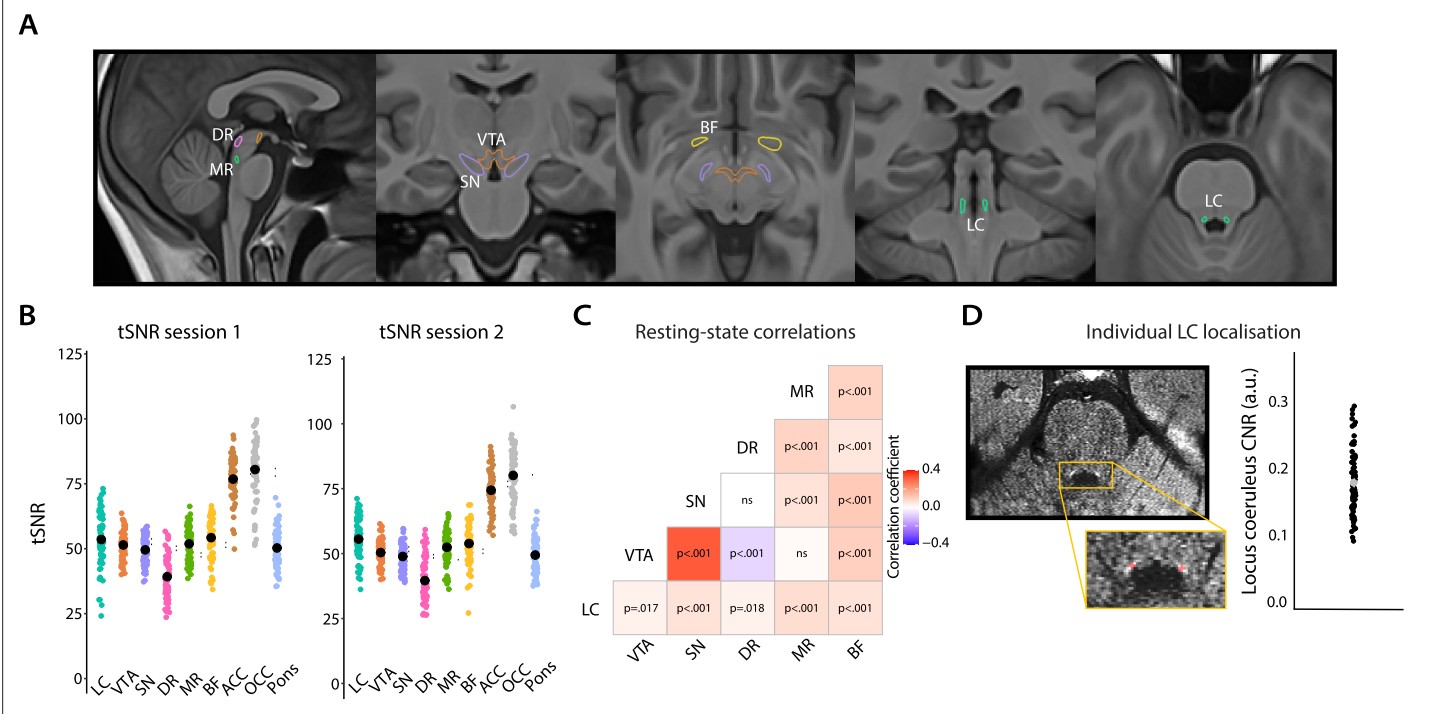

**Figure 2.** Overview of ROI definition and validation of the quality of subcortical fMRI data. (**A**) All subcortical ROIs overlaid on the group T1 template. (**B**) Individual data points showing the temporal signal-to-noise ratio for each ROI for session 1 (left) and session 2 (right; black points indicate the mean). (**C**) Correlation matrix showing that activity in subcortical nuclei co-varied positively with activity in other subcortical nuclei, with the strongest coupling present between the VTA and SN, which is to be expected given their close proximity, and the weakest (negative correlation) between the VTA and DR. Note: correlations were FDR-corrected and controlled for activity in the pons (n = 70). (**D**) FSE image of an example participant. Hyperintensities corresponding to the LC are visible in the yellow box (top). Using the FSE images, the LC (red) was manually delineated on the individual level following established protocols (*Clewett et al., 2016*; *Mather et al., 2017*). The graph shows the LC contrast-to-noise ratio for all participants (n = 67). The gray dot indicates the grand mean. LC, locus coeruleus; VTA, ventral tegmental area; SN, substantia nigra; DR, dorsal raphe; MR, medial raphe; BF, basal forebrain; ACC, anterior cingulate cortex; OCC, calcarine sulcus; CNR, contrast-to-noise ratio; ROI, region of interest; fMRI, functional magnetic resonance imaging; FDR, false discovery rate. Source data used to generate this figure is available in *Figure 2—source data 1*.

The online version of this article includes the following source data and figure supplement(s) for figure 2:

**Source data 1.** These csv files contains the data for *Figure 2B–D*.

**Figure supplement 1.** Overview of template building and coregistration steps performed in ANTs.

**Figure supplement 2.** Anatomical verification of accurate registration.

## Assessment of the quality of subcortical fMRI data

To assess the quality of the subcortical functional data we extracted the temporal signal-to-noise ratio (tSNR) from each ROI (see 'Materials and methods'). The average tSNR across the AAS ROIs (*Figure 2B*; range: 23.3–72.9) and cortical regions (range: 49.6–106.3) were in line with previous reports (*Brooks et al., 2013*, *Figure 1*: brainstem range: ~1–50 and cortex range ~50–112; *Singh et al., 2022*, *Figure 2B*: brainstem range: 0–50, cortex range: 0–50). We also replicated a recently reported pattern of positive (partial) correlations among the signal fluctuations in each pair of subcortical ROIs, controlled for activity in the pons (*Figure 2C*; *van den Brink et al., 2019*; see also *Singh et al., 2022*). Only the correlation between the DR and VTA was negative. Therefore, we are confident that the data had sufficient tSNR in our AAS ROIs to be able to assess pupil-AAS coupling.

## No pupil-AAS coupling using region- and participant-specific estimates of the HRF

The methods that have previously been used to examine co-fluctuations between pupil size and fMRI BOLD patterns worked under the assumption that the shape of the HRF during rest is homogeneous across the whole brain (*Breeden et al., 2017*; *DiNuzzo et al., 2019*; *Schneider et al., 2016*; *Yellin*

*et al., 2015*). This assumption may not be correct, and because even a 1 s latency difference between assumed HRF and actual HRF can have a significant impact on fMRI results (*Wall et al., 2009*), it may be important to account for regional and individual differences (*Bailes et al., 2023*; *Handwerker et al., 2004*) in the shape of the HRF. Indeed, subcortical structures have been characterized by faster BOLD responses (TTP 4–5 s; *Lau et al., 2011*; *Lewis et al., 2018*; *Yen et al., 2011*) compared to the cortex (TTP 5–6 s; *Lewis et al., 2018*; *Friston et al., 2000*). Therefore, we next estimated ROI-specific and participant-specific HRFs using an approach in which spontaneous pseudo-events were identified in our resting-state data and then aligned to determine the delay between the pseudo-events and corresponding BOLD signatures (*Rangaprakash et al., 2018*; *Wu et al., 2013*). The number of detected pseudo-events per ROI is shown in *Figure 3C*. Note that for some participants only one session was used to estimate these HRFs, so the number of detected pseudo-events for these participants tended to be smaller.

After carrying out pairwise comparisons, we found that, as expected, the TTP of the estimated HRFs was significantly faster for all subcortical AAS ROIs than for each of the two cortical ROIs ($M_{subcortical\ ROIs}$ = 4.7 s, $SD_{subcortical\ ROIs}$ = 0.6 s, $M_{cortical\ ROIs}$ = 5.4 s, $SD_{cortical\ ROIs}$ = 0.8 s; *Figure 3D*). The pupil-BOLD analysis using these specific HRFs, however, revealed no significant pupil-AAS correlations (*Figure 3G and H*). We only found that pupil size correlated negatively with activation in the OCC ($p_{corr}$<0.001 [pupil size], $p_{corr}$=0.018 [pupil derivative], false discovery rate [FDR]-corrected). Note that the negative sign of this correlation is consistent with what we reported above and with previous reports linking pupil size to decreased activity in the visual system (*Schneider et al., 2016*; *Yellin et al., 2015*).

In sum, the use of region- and participant-specific HRFs also did not result in significant pupil-AAS coupling. Importantly, by focusing on pseudo-events in the fMRI data, this approach still assumes that neurovascular coupling during rest (and other passive conditions) is characterized by the typical sluggish HRF used in event-related fMRI design.

## Positive pupil-AAS coupling using HRFs with rapid time-to-peaks

Since the analysis strategy so far unexpectedly did not result in pupil-AAS coupling, we let go of the assumption of a relatively slow HRF similar to that driving neurovascular coupling during task performance. Therefore, we further examined a potential relationship between pupil and AAS ROIs by examining how *systematically* varying the TTP (from 1 s to 6 s) of the default canonical HRF affected the coupling between pupil dynamics and our AAS ROIs. This analysis was inspired by a recent animal study (*Pais-Roldán et al., 2020*) showing that pupil-BOLD signal coupling dynamics vary across time.

We found that for almost all AAS ROIs the strength of pupil-BOLD coupling differed across TTPs (main effect of TTP; LC: p<0.001; VTA: p<0.001; SN: p<0.001; MR: p=0.043; BF: p=0.009, FDR-corrected for nine ROIs). The overall pattern shows that coupling between pupil size and AAS BOLD patterns increases with earlier TTPs. Specifically, we found positive correlations for all AAS regions at earlier TTPs (especially the 1 s [*Figure 4B*] and 2 s TTPs) but no significant correlations (LC, VTA, SN, DR, MR) at later TTPs (5–6 s; *Figure 4A*). For the OCC ROI, we found a positive correlation at the 1 s TTP, followed by a shift to negative correlations at later TTPs (4–6 s), which is in line with previous work (*Breeden et al., 2017*; *Schneider et al., 2016*; *Yellin et al., 2015*) and the results we reported above ('Whole-brain pupil-BOLD patterns consistent with previous studies'). The ACC correlated positively with pupil size at predominantly early TTPs (1–4 s; *Figure 4CD*), similar to the AAS ROIs, and in line with pupil-BOLD coupling in default-mode network areas (*Yellin et al., 2015*). To ensure the robustness of these findings, we repeated the analyses for each session separately. The pattern of results was visually and statistically similar between the two sessions, and in line with the results when both sessions were combined (*Figure 4—figure supplement 1*).

Similar analyses for the pupil derivative also showed significant differences in the strength of pupil-BOLD coupling across the TTPs for the VTA (p=0.012), SN (p=0.022), DR (p=0.002), ACC (p=0.009), and OCC (p<0.001; FDR-corrected for nine ROIs). The overall pattern and follow-up *t*-tests revealed similar but attenuated effects in comparison to pupil size (*Figure 4—figure supplement 2A*). The most prominent exception was the OCC, which showed a curvilinear relationship with largest correlations for intermediate TTPs (2–5 s), which is also visible upon inspecting the whole-brain maps in *Figure 4—figure supplement 2B*. The same analyses carried out for the control region in the pons revealed no main effect of TTP for pupil size or the pupil derivative, nor were there any positive or negative associations with pupil size or the pupil derivative for any TTP, attesting to the specificity of

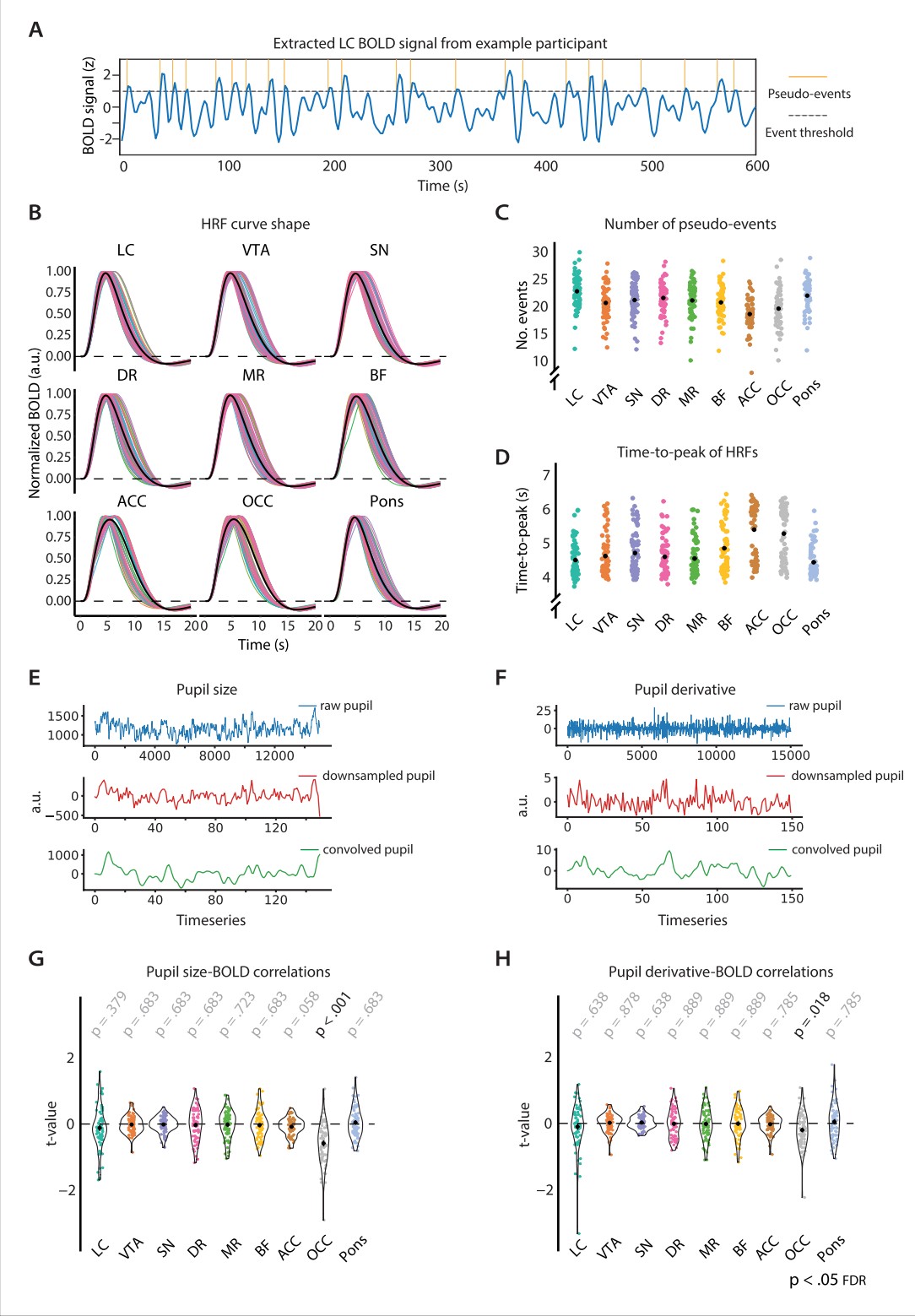

**Figure 3.** Overview of the analysis using region- and participant-specific estimates of the HRF. (**A**) One participant's pre-processed LC BOLD signal (concatenated across the two sessions), evaluated against a chosen threshold (>1 SD) to extract onsets of spontaneous neural events (indicated in yellow). (**B**) Estimated HRFs for each participant and ROI. Black lines indicate the average across participants. (**C**) The number of spontaneous neural events that were used to estimate the region- and participant-specific HRFs. Black dots indicate the mean. (**D**) TTP of the HRF for each participant and ROI. Black dots indicate the average across participants. Overview of the pupil data processing pipeline of one randomly chosen participant for pupil

*Figure 3 continued on next page*

*Figure 3 continued*

size (**E**) and the pupil derivative (**F**). The plots portray the raw pupil data (blue), the pupil time series down-sampled to the TR (red), and the convolved pupil time series (for LC HRF, green). The extracted *t*-values from the first-level resting-state HRF analysis for pupil size (**G**) and the pupil derivative (**H**). p-Values refer to one-sample *t*-tests (difference from zero; FDR-corrected; n = 70). LC, locus coeruleus; VTA, ventral tegmental area; SN, substantia nigra; DR, dorsal raphe; MR, median raphe; BF, basal forebrain; ACC, anterior cingulate cortex; OCC, calcarine sulcus; HRF, hemodynamic response function; BOLD, blood oxygen level-dependent; TTP, time-to-peak; FDR, false discovery rate. Source data used to generate this figure available in *Figure 3— source data 1*.

The online version of this article includes the following source data for figure 3:

**Source data 1.** These csv tables contain the data used for *Figure 3B–D, G–H*.

the pupil-BOLD associations found in our AAS and cortical ROIs. Statistical parametric maps including whole-brain results for all TTPs are shown in *Figure 4C* and *Figure 4—figure supplement 2B*.

Observing *Figure 4*, it seems that the LC exhibited greater inter-individual variability in pupil-BOLD signal correlations than most other ROIs. To explore the reasons for these differences, we investigated the correlation between a participant's *t*-statistic (i.e., for the LC at TTP = 1 s) and the size of their LC mask, as well as the tSNR in the LC. However, we did not find any significant correlation between the *t*-statistic and mask size (Spearman's rho(68) = 0.11, p=0.38) or tSNR (session 1: Spearman's rho(68) = 0.12, p=0.34; session 2: Spearman's rho(68) = –0.06, p=0.63). Additionally, we examined whether participants with a negative correlation in one session also exhibited a negative correlation in the other session, but this was not the case (Spearman's rho(54) = 0.18, p=0.19; n = 56 [participants with both sessions included]). Therefore, it remains an open question what drives these individual differences in pupil-LC BOLD signal correlations.

These exploratory analyses suggest that positive coupling between fluctuations in pupil diameter and AAS BOLD signal can be found when using a shorter transfer function (i.e., using TTPs of 1–2 s), but not with the broad HRFs that are typically used to model event-related BOLD responses (i.e., with a longer TTP; *Friston et al., 2000*).

## Positive pupil-AAS coupling when BOLD patterns closely follow pupil fluctuations

In our next analysis, we wanted to release the assumption that BOLD responses associated with pupil-size changes would resemble an HRF. Therefore, we correlated the BOLD signal from our AAS ROIs with the *unconvolved* pupil vector using a cross-correlation approach (*Pais-Roldán et al., 2020*; *Yellin et al., 2015*). This method also allowed us to interrogate the pupil-BOLD coupling in both time directions. To that end, we shifted the pupil vector 8 s backward and forward, in steps of 2 s, with negative lags (backward) corresponding to pupil changes preceding the BOLD signal and positive lags (forward) corresponding to pupil changes succeeding the BOLD signal (*Figure 5A and B*).

Critically, and in line with the TTP analysis, we found significant positive pupil-BOLD correlations for all AAS ROIs (except the DR), with the strongest correlations occurring at lag 0 (*Figure 5A*). These results again suggest that the relationship between pupil size and AAS activity is temporally close, rather than following the shape of an HRF with a 5 or 6 s TTP. In addition, these patterns of results appeared to be stable across the two sessions (*Figure 5—figure supplement 1*). For the pupil derivative (*Figure 5—figure supplement 2*), we observed a similar pattern in SN and VTA, with stronger correlations at lag 0, although overall the correlation coefficients were attenuated compared to those for pupil size, or not present in some AAS ROIs (LC, MR, and BF).

For comparison, we also computed cross-correlations between pupil size and BOLD signal extracted from our validation and control ROIs (ACC, OCC, pons; *Figure 5B*). The OCC showed strong negative correlations at lags +4 to +8 s, similar to previous studies (*Murphy et al., 2014*; *Schneider et al., 2016*), and in line with the replication and TTP analyses reported above. However, the OCC also showed a positive correlation with pupil size at short lags (–2 s to +2 s). Similarly, we found that both ACC and OCC correlated most strongly with the pupil derivative (*Figure 5—figure supplement 2*) at relatively short positive lags (0 s to +4 s), with a shift to a strong negative correlation at maximum positive lags (+8 s), especially in the OCC.

To determine whether the observed pupil-AAS BOLD signal cross-correlation results were independent of the LC, we recomputed the correlations between pupil size and BOLD signal after partialing out the contribution of the LC BOLD signal using linear regression. As shown in *Figure 5C*, partialing

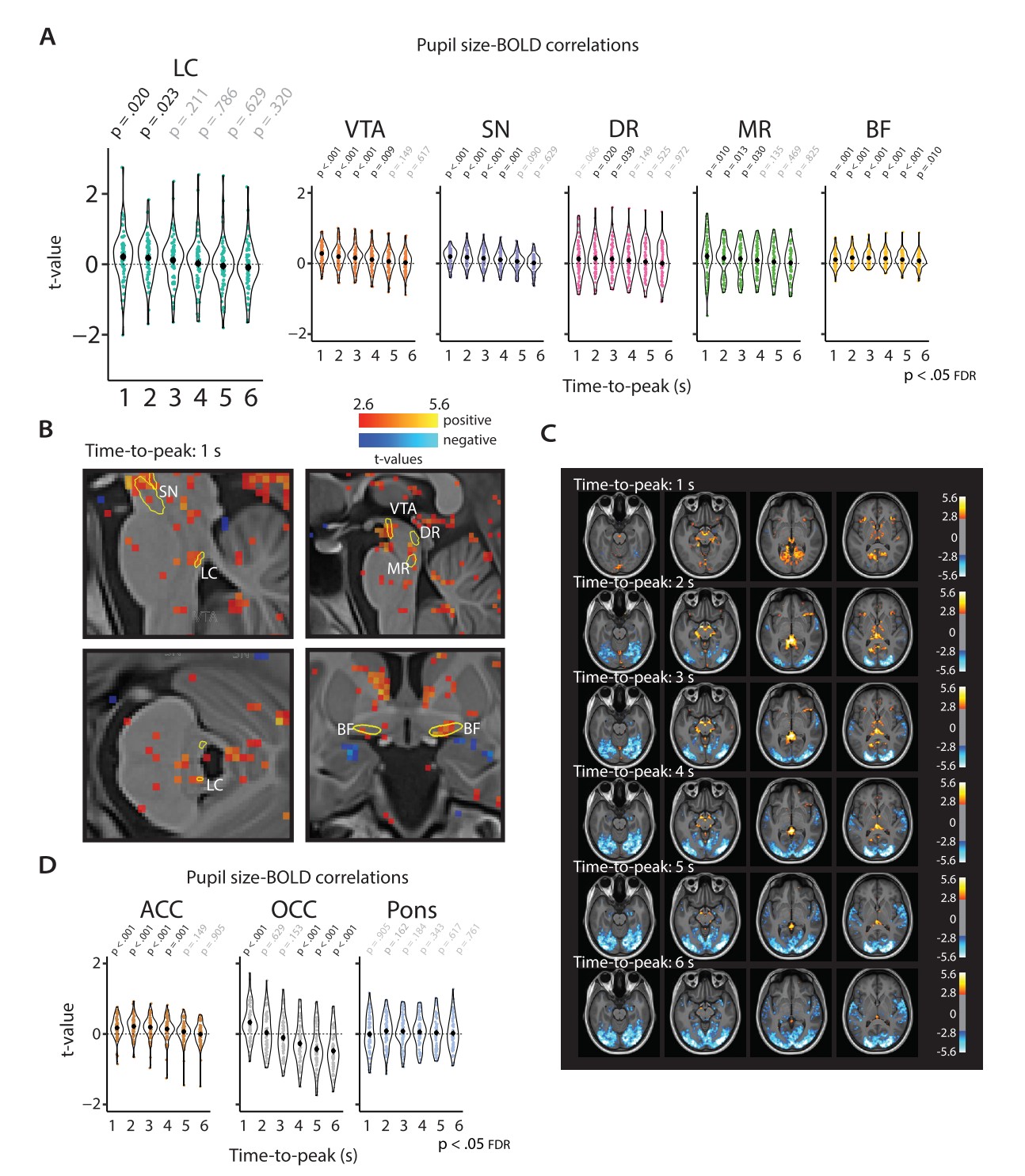

**Figure 4.** Pupil-BOLD correlations based on systematic adjustment of the TTP of the HRF. Graphs show the extracted *t*-statistics as a function of the systematically adjusted TTP for each participant in the subcortical ROIs (**A**) and the validation and control ROIs (**D**). p-Values refer to one-sample *t*-tests (difference from zero; FDR-corrected). Black dots indicate the mean. (**B**) Statistical maps showing pupil-BOLD correlations in the subcortex for the 1 s TTP. (**c**) Statistical maps showing unsmoothed pupil-BOLD correlations across the cortex for each TTP (1–6 s). All statistical maps were thresholded at p<0.005 (uncorrected) for visualization purposes only (n = 70). LC, locus coeruleus; VTA, ventral tegmental area; SN, substantia nigra; DR, dorsal raphe; MR, median raphe; BF, basal forebrain; ACC, anterior cingulate cortex; OCC, calcarine sulcus; BOLD, blood oxygen level-dependent; FDR, false discovery rate; TTP, time-to-peak; HRF, hemodynamic response function; ROI, region of interest. Source data used to generate this figure is available in *Figure 4—source data 1*.

*Figure 4 continued on next page*

*Figure 4 continued*

The online version of this article includes the following source data and figure supplement(s) for figure 4:

**Source data 1.** This folder contains the csv files and statistical maps used to make *Figure 4*.

**Source data 2.** This contains the csv tables used in *Figure 4—figure supplement 1*.

**Source data 3.** This folder contains the csv tables and statistical maps used in *Figure 4—figure supplement 2*.

**Figure supplement 1.** Pupil size-BOLD correlations based on systematic adjustment of the TTP of the HRF for both sessions separately.

**Figure supplement 2.** Pupil derivative-BOLD correlations based on systematic adjustment of the TTP of the HRF.

out the contribution of the LC resulted in a slight but significant drop for the BF (p=0.028). The signal from the VTA, SN, MR, and BF remained significantly correlated with pupil size (ps<0.030). These results contribute further evidence to the notion that pupil size is not selectively driven by the LC, but that a broader network of AAS nuclei is involved.

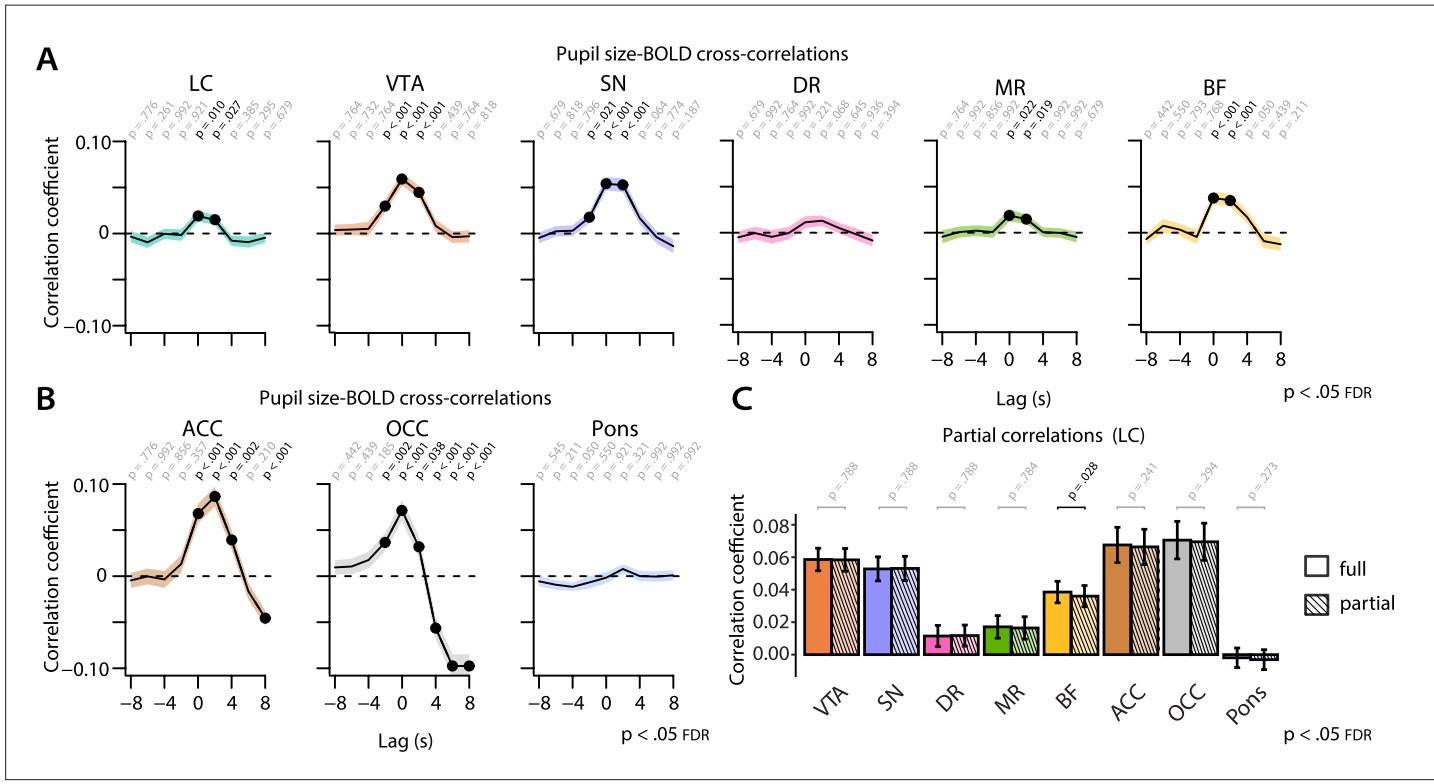

**Figure 5.** Cross-correlations between the unconvolved pupil time series and the BOLD time series at various time lags. Pupil-BOLD time-series cross-correlations for all AAS nuclei ROIs (**A**) and control ROIs (**B**). Negative (positive) time lags indicate that the pupil signal precedes (follows) the BOLD signal. p-Values refer to one-sample *t*-tests for the corresponding time bin. (**C**) Statistical comparisons of pupil size-BOLD signal coupling at lag = 0 s before (full) and after regressing out LC BOLD signal (partial). Statistics refer to two-tailed *t*-tests (FDR-corrected) comparing the full and partial correlation coefficients. Black lines (**A, B**) and bar plots (**C**) indicate the grand mean and shaded regions (**A, B**) and error bars (**C**) indicate ± standard error of the mean (SEM). Black font and black dots indicate significant time bins (p<0.05, FDR-corrected) (n = 70). BOLD, blood oxygen level-dependent; AAS, ascending arousal system; ROI, region of interest; LC, locus coeruleus; FDR, false discovery rate. Source data used to generate this figure is available in *Figure 5—source data 1*.

The online version of this article includes the following source data and figure supplement(s) for figure 5:

**Source data 1.** These csv tables contain the data used for *Figure 5*.

**Source data 2.** This contains the csv files used in *Figure 5—figure supplement 2*.

**Figure supplement 1.** Cross-correlations between the unconvolved pupil size time series and the BOLD time series at various time lags for each resting-state session.

**Figure supplement 2.** Cross-correlations between the unconvolved pupil derivative time series and the BOLD time series at various time lags.

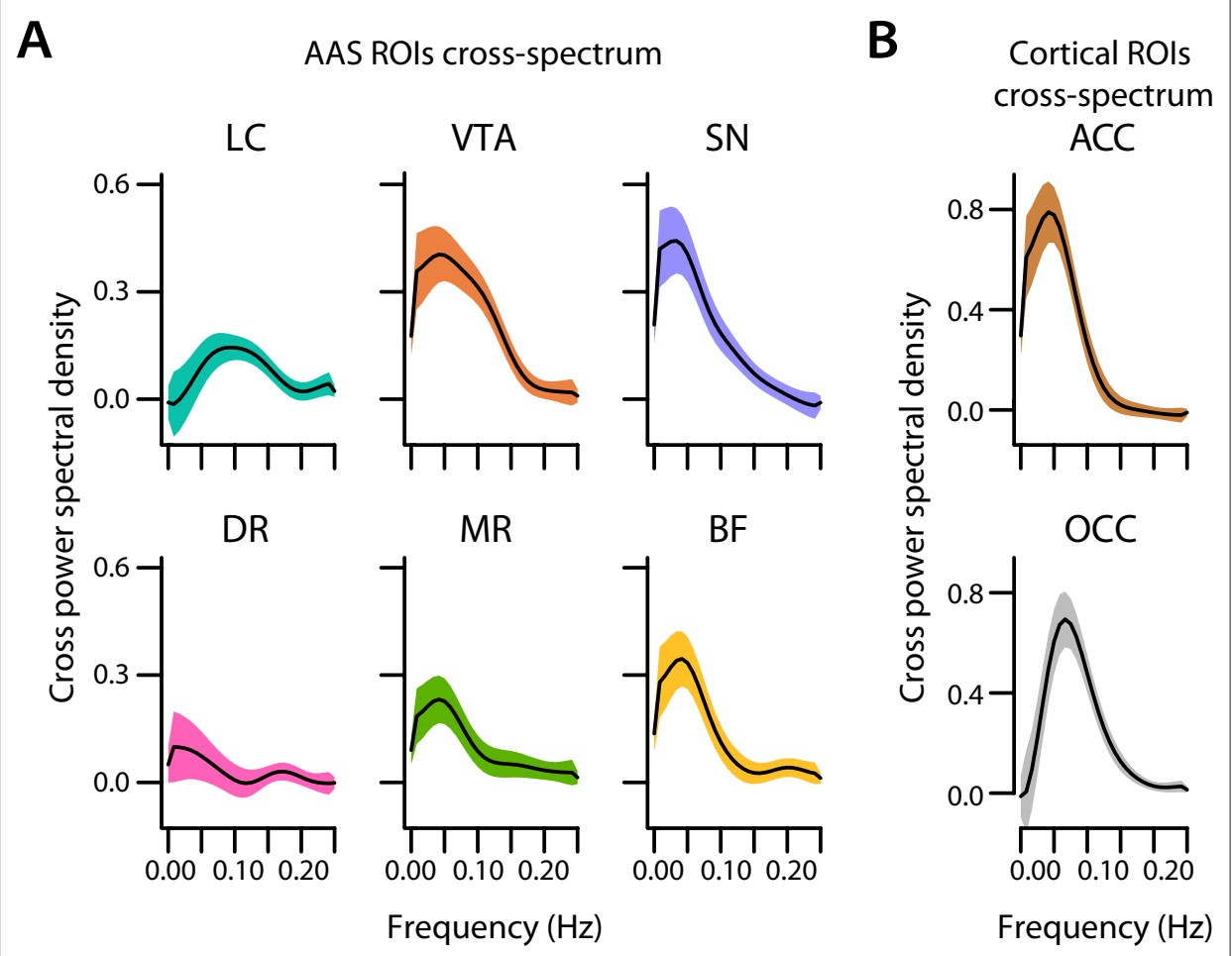

**Figure 6.** Cross-spectral power density analysis. Cross-spectral power density for subcortical ROIs (**A**) and cortical ROIs (**B**) averaged across participants. Black lines indicate the grand mean and shaded regions indicate ± SEM (n = 70). LC, locus coeruleus; VTA, ventral tegmental area; SN, substantia nigra; DR, dorsal raphe; MR, median raphe; BF, basal forebrain; ACC, anterior cingulate cortex; OCC, calcarine sulcus; ROI, region of interest. Source data used to generate this figure is available in *Figure 6—source data 1*.

The online version of this article includes the following source data for figure 6:

**Source data 1.** This folder contains the csv tables used in *Figure 6*.

Together, the TTP analysis and cross-correlation analysis yield essentially the same outcome, suggesting that no HRF convolution is needed to characterize the relationship between pupil size and AAS BOLD patterns during rest.

## Pupil-AAS coupling is largely driven by oscillations in low-frequency band

Finally, to better understand the nature of the pupil-AAS coupling, we carried out an exploratory cross-spectral density analysis (*Figure 6A, B*). The cross-power spectral density is the Fourier transform of the cross-correlation functions reported above, and hence expresses the relationship between the pupil and AAS signals in the frequency domain. To determine which frequency bands were driving the observed positive pupil-AAS correlations, we calculated the cross-spectral power density (see 'Materials and methods') of the pupil size time series and average BOLD time series extracted from each ROI. A simple peak detection indicated that the correlations for most AAS nuclei and both cortical ROIs (ACC and OCC) were largely driven by frequencies between 0.04 and 0.09 Hz (LC: 0.09; VTA: 0.04; SN: 0.03; DR: 0.008; MR: 0.04; BF: 0.04; ACC: 0.04; OCC: 0.07; *Figure 6AB*).

## Discussion

In this study, we examined whether, during rest, non-luminance-related spontaneous fluctuations in pupil size were associated with fluctuations in BOLD signal in nuclei part of the AAS. We found a positive correlation between pupil size and BOLD signal in all of the AAS ROIs: LC, VTA, SN, DR, MR, and (sublenticular) BF. This finding is in line with recent animal studies (*Joshi et al., 2016*; *Reimer et al., 2016*) indicating that pupil changes reflect activity in multiple neuromodulatory systems, not only the noradrenergic system. Critically, using two different methodological approaches, we found that pupil-AAS coupling was strongest when the two signals were assumed to occur close in time. This means that during rest, unlike in response to task events (*de Gee et al., 2017*), BOLD signal fluctuations in AAS nuclei immediately follow fluctuations in pupil size. This correlation was largely driven by slow oscillations (i.e., ~0.05–0.1 Hz) in both measures. Together, our results suggest that pupil size can be used as a noninvasive readout of AAS activity and reveal new insights into the temporal dynamics of pupil-AAS coupling during rest.

We found robust positive correlations between pupil size and BOLD signal in five of our AAS ROIs: LC, VTA, SN, MR, and (sublenticular) BF. The correlations with the VTA, SN, MR, and BF BOLD signals remained significant after controlling for the effect of the LC BOLD signal, suggesting that these AAS nuclei have *independent* influences on pupil size. The positive relationship with pupil size was less robust for the DR, and only significant in the TTP analyses, perhaps because this area had lower tSNR than the other subcortical ROIs. Thus, unlike what findings from previous studies (e.g., *Murphy et al., 2014*) may have implicitly suggested, the coupling between pupil size and activation of the AAS is not specific for the LC. Our findings contribute to a growing body of literature showing a more general role for AAS nuclei in driving pupil size changes. Specifically, our findings are consistent with recent animal studies that found co-fluctuations in pupil size and LC and BF activity during rest (*Joshi et al., 2016*; *Reimer et al., 2016*); with studies showing that optogenetic activation of the LC or DR increases pupil size (*Breton-Provencher and Sur, 2019*; *Cazettes et al., 2021*); and with human task-related fMRI work showing positive correlations between event-related pupil responses and BOLD responses in the LC, VTA, and BF (*de Gee et al., 2017*). Unfortunately, it has become common practice for researchers to interpret task-related pupillometry data in terms of the role of the LC in cognitive and brain function. However, our findings reinforce previous arguments (*Joshi and Gold, 2020*) that changes in pupil size should not be used to infer a selective role for the LC.

The temporal relationship between pupil size and AAS BOLD response patterns was different than we had expected based on reports from previous resting-state and event-related fMRI studies (*de Gee et al., 2017*; *Murphy et al., 2014*). Namely, we found that BOLD signal in the AAS closely followed the corresponding pupil fluctuations. These results were corroborated by a series of analyses in which we convolved the pupil time series with HRFs with systematically varied TTP (1–6 s). Pupil time series that were convolved with the canonical HRF (TTP = 6 s) or region-specific HRFs based on a point process approach were not significantly related to AAS activation. Instead, maximal and significant pupil-AAS coupling was found using HRFs with TTPs of 1–3 s. In addition, cross-correlations between the unconvolved pupil time series and AAS BOLD time series similarly revealed maximum correlations when the signals occurred close in time (at lags of 0–2 s). For the pupil derivative, we obtained similar results, with strongest cross-correlations around lag 0 s to +2 s, although the correlation coefficients were overall attenuated and only remained significant in the VTA and SN. Our findings are in line with previous research that has suggested that subcortical regions (*Lewis et al., 2018*) and AAS nuclei (*de Gee et al., 2017*) are characterized by faster event-related hemodynamic responses than cortical regions. However, our findings suggest an even closer temporal relationship between pupil size and BOLD response patterns in AAS nuclei during rest. Therefore, these findings, although correlational, have implications for our understanding of the time scale at which AAS regions might drive changes in pupil diameter.

Furthermore, our findings may provide a reason why most previous human resting-state pupil-fMRI studies (e.g., *Breeden et al., 2017*; *Schneider et al., 2016*; *Yellin et al., 2015*) did not find or report pupil-AAS coupling. Namely, these studies only investigated pupil-BOLD coupling with longer time lags between pupil changes and corresponding BOLD response patterns. Using standard TTPs, we (broadly) replicated previously reported associations between pupil size and cortical BOLD response patterns (e.g., negative coupling with the visual cortex, positive coupling with the thalamus and posterior cingulate cortex; e.g., *Murphy et al., 2014*; *Schneider et al., 2016*; *Yellin et al., 2015*). However,

we did not find evidence that standard TTPs characterized the coupling between resting-state pupil size and AAS activation. Although one of these previous studies did report pupil-LC coupling (*Murphy et al., 2014*), we were unable to replicate this finding in our data, despite using the same convolution methods, LC mask, and other methodological details of this study, and despite a much larger sample size (70 vs. 14 included participants). Although we cannot make a direct comparison between the shape of human and mouse HRFs, a recent study in rats (*Pais-Roldán et al., 2020*) also found temporally close positive coupling (i.e., 1 s TTP) between resting-state pupil dynamics and BOLD signal in specific areas in the brainstem including the A5 noradrenergic cell group (which projects to the spinal cord). This study, however, did not find evidence for a coupling between pupil dynamics and any of the AAS nuclei. Although this seems inconsistent with our findings, note that the rats in this study were anesthetized. Preliminary evidence suggests that behavioral states can strongly modulate pupil-AAS coupling (*Megemont et al., 2022*), and therefore it is possible that in an anesthetized state pupil-AAS coupling is reduced or absent.

Our findings raise the question *how* the neuronal activity in AAS nuclei that contributes to resting-state pupil fluctuations is coupled to the BOLD signal in these areas. The main frequency band that drove our pupil-AAS coupling was ~0.05–0.1 Hz. Interestingly, recent work in the mouse cortex has shown that the ultra-slow (~0.1 Hz) BOLD fluctuations that are characteristic of resting-state fMRI data are entrained by ultra-slow vasomotor oscillations that lead to rhythmic changes in the diameter of brain arterioles. These vasomotor oscillations, in turn, are entrained by rhythmic local neuronal activity in the same ultra-low-frequency band (*Drew et al., 2020*; *Mateo et al., 2017*). These findings beg the question whether this neurovascular coupling sequence may be responsible for our findings. However, in the mouse brain this sequence, from neuronal activity and vasomotion to blood oxygenation levels, was estimated to last approximately 2.6 s (*Mateo et al., 2017*). This seems inconsistent with the 0–2 s interval between our estimated timing of the AAS neuronal activity underlying pupil fluctuations and the timing of corresponding AAS BOLD signals. Future work in animal models should therefore examine the physiological basis of these seemingly close temporal relationships between activity of AAS nuclei and corresponding changes in pupil size by simultaneously measuring pupil size changes and rhythmic BOLD fluctuations during awake rest.

Our study has several potential limitations. First, although our EPI sequence had a higher spatial resolution (2 mm isotropic) than previous studies linking pupil size to BOLD (e.g., 3.5 mm isotropic in *Murphy et al., 2014*), imaging small subcortical structures at this conventional spatial resolution may have led to partial-volume averaging (*Forstmann et al., 2016*; *Liu et al., 2017*), especially in the LC, the smallest of our ROIs. To mitigate this concern, we did not apply spatial smoothing to the EPI data. Our confidence in the LC imaging data reported here is also bolstered by the fact that the LC showed the same pattern of results as other, much larger nuclei, including the VTA, SN, and BF, that are less susceptible to partial-volume averaging effects. We also note that a further increase in spatial resolution at 3T would be accompanied by a dramatic drop in signal-to-noise ratio (*Murphy et al., 2007*), and therefore would not per se lead to a better signal from the AAS regions. Although future studies combining pupillometry with ultra-high-field fMRI (e.g., 7T) could circumvent this problem, measuring pupil size in ultra-high-field scanners is still challenging. A second potential limitation concerns the proximity of some of the subcortical ROIs to air- and cerebrospinal fluid-filled cavities as well as major arteries, making them particularly prone to movement and other sources of physiological noise (*Brooks et al., 2013*). To mitigate this concern, we included an extensive physiological noise correction model, accounting for measured cardiac and respiratory signal components as well as residual signal from the fourth ventricle. Furthermore, if the BOLD signal in AAS nuclei was largely driven by noise, this could not explain the robust temporal relationship between AAS BOLD and pupil size, and the selective absence of pupil-BOLD coupling in the pons, our control region that is also susceptible to physiological noise artifacts. A third drawback is that our analyses were limited by the temporal resolution of our fMRI data. Due to our 2 s TR, we were unable to interrogate potentially meaningful, faster pupil-BOLD correlations. Future studies using ultra-high-field fMRI and/or simultaneous imaging techniques (*Barth et al., 2016*; *Lewis et al., 2016*) can speed up image acquisition and assess the presence of pupil-BOLD correlations at a faster timescale.

In conclusion, we show that spontaneous changes in pupil size that occur during rest reflect activity in a variety of nuclei that are part of the AAS. This suggests that pupil size can be used as a noninvasive and general index of AAS activity, in contrast to previous work suggesting a selective role for the LC

in arousal-related pupil size changes. However, the nature of pupil-AAS coupling during rest appears to be vastly different from task-related pupil-AAS coupling, which has previously been modeled using a canonical HRF. Together, our findings provide new insights into the nature and temporal dynamics of AAS-linked pupil size fluctuations.

# Materials and methods

This study was preregistered on the Open Science Framework before data analysis (https://osf.io/5rjcf/). Note that preregistration occurred after data collection; due to circumstances surrounding the global pandemic, already collected data was used to address our hypotheses. As we could not replicate previous findings, we needed to deviate from the preregistration. When we deviated from the preregistration, this will be explicitly mentioned below.

## Participants

Seventy-four right-handed participants were recruited from New York University and completed two resting-state sessions on two consecutive days. Two participants were excluded entirely and one session for one participant was excluded due to technical issues with the scanner, leaving a sample size of 72 participants for the current study (mean age: 22.5 y; age range: 18–33 y; sex: 39 females, 33 males; gender: 39 females, 33 males, ethnicity: Asian/Asian American: 23, Asian: 1, Black/African American: 2, Caucasian/White/European American: 23, Hispanic non-white: 9, Latin: 1, Middle Eastern: 1, Mixed: 6, unknown: 6). The resting-state and pupil data were collected in the context of a larger study on extinction learning not published yet. On both days, the resting-state sessions were collected before the participants carried out a separate event-related fMRI task. The sample size (n = 72) was determined based on a between-group difference within the task that was conducted following the resting-state session used in this study. Exclusion criteria for participation were as follows: current treatment or treatment in the last year of psychiatric, neurological, or endocrine disease, current treatment with any medication, average use of >3 alcoholic beverages daily, average use of recreational drugs, habitual smoking, uncorrected vision, and contraindications for MRI. The two resting-state sessions were part of a larger study of which the data will not be reported here. The study was approved by the University Committee on Activities Involving Human Subjects at New York University (Institutional Review Board #2016-2), and the study was conducted in accordance with these guidelines and regulations. All participants provided written informed consent. Participants received a payment ($35 per hour) for their participation.

## Procedure

All participants completed two resting-state sessions of 5 min each, 24 hr apart (±2 hr). During the session, they were instructed to think of nothing in particular, let their mind wander but not to have any repetitive thoughts such as counting. They were instructed to keep their eyes open and maintain their gaze on a centrally presented fixation dot (RGB: 60, 60, 255) on a gray screen (RGB: 125, 125, 125).

## MRI data acquisition

MRI data was acquired using a Siemens MAGNETOM Prisma 3T MR scanner. T2*-weighted BOLD images were recorded using a customized multi-echo EPI sequence (2.0 mm isotropic) with ascending slice acquisition (58 axial slices; TR = 2 s; TE = 14.4, 39.1 ms; partial Fourier = 6/8; GRAPPA acceleration factor = 2; multiband acceleration factor = 2; flip angle = 65°; slice matrix size 104 × ×104; slice thickness = 2.0 mm; FOV: 208 × 208 mm; slice gap = 0; bandwidth: 2090 Hz/px; echo spacing: 0.56 ms). Multi-echo EPI protocols can be used to avoid the tradeoff between BOLD sensitivity in the cortex and subcortex (*Turker et al., 2021*). To account for regional variation in susceptibility-induced signal dropout, voxel-wise weighted sums of both echoes were calculated based on local contrast-to-noise ratio (*Poser et al., 2006*). A structural image (0.9 mm isotropic) was acquired using a T1-weighted 3D MP-RAGE (TR = 2.3 s; TE = 2.32 ms; flip angle = 8°, FOV = 256 × 256 × 230 mm). A fast-spin echo (FSE) neuromelanin-sensitive structural scan was acquired for delineation of the LC (11 axial slices, TR = 750 ms, TE = 10 ms, flip angle = 120°, bandwidth = 220 Hz/Px, slice thickness = 2.5 mm, slice gap = 3.5 mm; in-plane resolution = 0.429 × 0.429 mm). Note that a large slice gap is a common feature

in the use of FSE scans for LC imaging (*Liu et al., 2017*). This procedure allows for a high in-plane resolution but with a thicker slice thickness, resulting in elongated voxels that match the cylindrical shape of the LC. To minimize excessive movement during scanning, we secured participants' heads in a pillow and medical tape was attached across their foreheads to provide immediate tactile feedback in case of any movement, which has been shown to reduce motion (*Krause et al., 2019*).

## MRI data preprocessing

Preprocessing of MRI data was carried out using Advanced Normalization Tools v2.1 (ANTs) and SPM12 (https://www.fil.ion.ucl.ac.uk/spm; Wellcome Department of Imaging Neuroscience, London, UK). Here, we deviated from the preregistration, as we reported we would carry out all preprocessing steps in SPM12. Since ANTs SyN was found to be the best performing method for normalization (*Ewert et al., 2019*), some steps, including standardization and registration, were carried out using ANTs instead.

A whole-brain group template ($T1_{template}$) was generated using all MP-RAGE scans (using antsMult ivariateTemplateConstructinon2.sh; *Figure 2—figure supplement 1A*). This process involved two steps: (1) participants' whole-brain T1 scans ($T1_{native\ space}$) were coregistered to a common group space ($T1_{group\ space}$); then (2) these coregistered scans were averaged to form a whole-brain group template ($T1_{template}$). An initial fixed image was created by averaging all input files. For registration (or 'normalization') of input images, a set of linear (rigid, then affine) and nonlinear (SyN) algorithms were used. Each nonlinear registration was performed over four levels of increasingly fine-grained resolutions ($100 \times 70 \times 50 \times 10$ iterations). We applied an N4 bias field correction on moving images before each registration (using *N4BiasFieldCorrection* function). Cross-correlation was the similarity metric used for registration. Greedy SyN (SyN) was the transformation model used for registration. The gradient step size for refining template updates was set at 0.20 mm. After the whole-brain template image ($T1_{template}$) was generated, for optimal coregistration, all individual MP-RAGE scans ($T1_{native\ space}$) were submitted to a new coregistration step (using antsRegistration.sh; *Figure 2—figure supplement 1B*). For this we performed linear (rigid, then affine), followed by nonlinear (SyN), registration steps, resulting in optimized individual whole-brain scans in template space ($T1_{group\ space}$). To ensure that individual MP-RAGE scans and the group template ($T1_{native\ space}$ to $T1_{group\ space}$) had been registered accurately, we carried out visual checks for each individual using anatomical boundaries and ventricles as reference regions. We also calculated the average (*Figure 2—figure supplement 2B*) and standard deviation (*Figure 2—figure supplement 2C*) of the individual warped T1 images (n = 72). On the individual T1s standard deviation image (*Figure 2—figure supplement 2C*), light areas (high values) indicate where the coregistration is more variable, while dark areas (low values) indicate low variability across the T1s. Overall, these images show that the brainstem/subcortical areas are well-registered (i.e., dark areas in the standard deviation image).

Mutual information maximization-based rigid-body registration was used to register MP-RAGE scans and functional images. Functional images were motion-corrected using rigid-body transformations. To move the functional images into group space, the affine transforms and displacement field transformations from the final coregistration ($T1_{native\ space}$ to $T1_{group\ space}$) for each participant were applied to their respective functional images (using linear interpolation). To avoid contamination of AAS BOLD activity by signal from adjacent structures, all analyses reported in this article, except those aimed at replicating previous studies (see section 'Comparisons with previous studies'), were performed without spatial smoothing.

We applied a movement and physiological noise correction model with 33 regressors in SPM12. These included six movement parameter regressors (three translations, three rotations) derived from rigid-body motion correction, high-pass filtering (1/128 Hz cut-off) and AR(1) serial autocorrelation corrections. In addition these included retrospective image-based correction (RETROICOR) of physiological noise artifacts (*Glover et al., 2000*) regressors. Raw pulse was preprocessed using PulseCor (https://github.com/lindvoo/PulseCor; *de Voogd, 2023a*) implemented in Python for artifact correction and peak detection. Fifth-order Fourier models of the cardiac and respiratory phase-related modulation of the BOLD signal were specified (*van Buuren et al., 2009*), yielding 10 nuisance regressors for cardiac noise and 10 for respiratory noise. Additional regressors were calculated for heart rate frequency, heart rate variability, (raw) abdominal circumference, respiratory frequency, respiratory amplitude, and respiration volume per unit time (*Birn et al., 2006*), yielding a total of 26 RETROICOR

regressors (https://github.com/can-lab/RETROICORplus; *Krause, 2021*). An additional regressor was added to remove signal fluctuations from the fourth ventricle, which was manually delineated using individual MP-RAGE scans (M = 90, SD = 34 voxels). This movement and physiological noise correction model was added to all general linear models (GLMs) described below. Note that since scrubbing did not affect the outcomes of our initial preregistered analysis (replication of *Murphy et al., 2014*), we report only the analysis without scrubbing and opted to refrain from applying scrubbing in the subsequent exploratory analyses.

## Pupil data acquisition and preprocessing

Pupil size was recorded using an MR-compatible eye-tracker (EyeLink 1000 Plus; SR Research, Osgoode, ON, Canada) at a sampling rate of 250 Hz. The eye-tracker was placed at the end of the scanner bore, such that the participant's right eye could be tracked via the head coil mirror. Before the start of each resting-state session, we began with a calibration of the eye-tracker using the standard five-point EyeLink calibration procedure.

Moments when the eye-tracker received no pupil signal (i.e., during eye blinks) were marked automatically during acquisition by the manufacturer's blink detection algorithm. Pupil data was preprocessed using PupCor (https://github.com/lindvoo/PupCor; *de Voogd, 2023b*) implemented in Python. Missing and invalid data due to blinks were replaced using linear interpolation for the period from 100 ms before blink onset to 400 ms after blink offset. Following the automated interpolation procedure, the data were manually checked and corrected if any artifacts had not been successfully removed. Two sessions from two separate participants were excluded due to technical problems with the eye-tracker. To ensure the pupil data was of good quality, a session was excluded from all analyses if the raw pupil data contained >25% invalid samples (marked automatically during data acquisition by EyeLink's blink detection algorithm; *n* sessions *excluded* = 15). For the remaining sessions (*n* sessions *included* = 126; *n* participants = 70, average proportion invalid samples = 6.4%), we computed pupil size as well as the first-order derivative of the pupil size time series. The latter describes the slope of changes in pupil size, where positive values reflect pupil dilation and negative values reflect pupil constriction. Because pupil size lags behind the underlying neural activity in AAS nuclei (including the LC and DR; *Cazettes et al., 2021*; *Joshi et al., 2016*; *Liu et al., 2017*; *Reimer et al., 2016*), and in line with previous neuroimaging studies (*Pfeffer et al., 2022*; *Schneider et al., 2016*; *Yellin et al., 2015*), we shifted the pupil time series 1 s back in time. This step was only omitted when we attempted to replicate *Murphy et al., 2014*. Both pupil time series (i.e., pupil size and pupil derivative) were then resampled to the TR (2 s) resolution (0.5 Hz). To detect further artifactual samples, within each 2 s time bin, any sample ±3 SD outside the time bin mean was removed, after which the average of the corresponding time bin was recalculated from the remaining non-artifactual samples (percentage samples recalculated = 0.04%; as in *Murphy et al., 2014*). The results of these preprocessing steps were two pupil time series (pupil size and pupil derivative) that were equal in length to the number of fMRI volumes (i.e., 150) collected in each session.

## Definition of ROIs

The LC was delineated on each participant's FSE scan using ITK-SNAP (version 3.8.0; *Yushkevich et al., 2006*). Two raters (BL and a research assistant) manually identified LC voxels following established protocols (*Clewett et al., 2016*; *Mather et al., 2017*). Pairwise dice similarity coefficients between both raters were high (M = 0.96; range: 0.70–1.00). As described in *Clewett et al., 2016*, left and right LC regions were identified in the axial slice ~7 mm below the inferior colliculus. Within this slice, two regions were delineated in the form of a cross ~1.29 mm wide and ~1.29 mm high (3 × 3 voxels, see *Figure 2D*), covering the 1–2 mm of LC neurons in this slice. The center voxel for each cross was placed on the voxel with the highest signal intensity that fell within an area anatomically consistent with the location of the LC. If the peak voxel was located immediately adjacent to the fourth ventricle, the center of the ROI was placed one voxel further away from the ventricle. This ensured that the peak voxel but no fourth ventricle voxels were included in the ROI. To ensure we captured LC intensity signal, we calculated an objective measure for comparisons namely contrast-to-noise ratios between the average signal intensity in the LC relative to a pontine reference region. Contrast-to-noise ratios were positive for all participants, indicating that LC intensity was consistently higher than pontine intensity (M = 0.17; range: 0.08–0.29; *Figure 2D*). The

obtained contrast-to-noise ratios were in line with previous reports (*Clewett et al., 2016*; *Mather et al., 2017*). For three participants, we could not delineate LC ROIs because movement led to poor-quality FSE images. For these participants, we used the group LC ROI (i.e., average of all individual LC ROIs in group space thresholded at 2 SD above the mean). This group LC ROI was also used for visualization purposes. In line with current standards (*Yi et al., 2021*), the LC masks were then transformed (using nearest-neighbor interpolation) into template space by applying the linear and nonlinear transformations from the final coregistration ($T1_{native space}$ to $T1_{group space}$) and resliced, resulting in the final individual LC masks in functional space (range of size in functional space: 1–7 voxels, M = 3.4, SD = 1.3 voxels).

Published probabilistic atlases were used for the remaining subcortical ROIs as there are no established protocols for individual segmentation of these regions: VTA (*Trutti et al., 2021*; size in functional space = 45 voxels), SN (*Alkemade et al., 2020*; 84 voxels), DR (*Beliveau et al., 2015*; 4 voxels), MR (*Beliveau et al., 2015*; 3 voxels), and the sublenticular (Ch4) part of the BF (*Eickhoff et al., 2005*; *Zaborszky et al., 2008*; 31 voxels), which includes the cholinergic nucleus basalis of Meynert (see *Figure 2A*). All atlases were originally in MNI space. To move the atlases into our study-specific template space, antsRegistration.sh (using the same parameters as described above) was applied to generate the transformation matrices between MNI space and template space, which were applied to each ROI mask. Each subcortical mask was then thresholded and resliced to the functional space. We visually verified anatomical accuracy of the ROIs on the T1s following published guidelines (*Beliveau et al., 2015*; *Trutti et al., 2021*) and by visually inspecting the original MNI-space ROIs and our transformed ROIs (*Alkemade et al., 2020*; *Zaborszky et al., 2008*). In the preregistration, we reported that we would only be examining pupil-BOLD coupling in the LC, VTA, and SN. We also opted to include the raphe nuclei and BF since recent evidence shows that they are involved in driving pupil size during task behaviors (*Cazettes et al., 2021*; *de Gee et al., 2017*).

Previous studies (*Schneider et al., 2016*; *Yellin et al., 2015*) have found a robust relationship between pupil size and BOLD patterns in the occipital cortex (OCC) and anterior cingulate cortex (ACC). Therefore, we included these two cortical regions as additional validation ROIs. Specifically, we obtained masks of the calcarine sulcus in the OCC (size in functional space = 1975 voxels) and ACC (1148 voxels) using the automated anatomical labeling atlas in SPM (*Tzourio-Mazoyer et al., 2002*). Lastly, to explore the specificity of our pupil-AAS BOLD results, we delineated a cubic ROI in the medial part of the basis pontis (pons; 8 voxels), which served as a control region in which we did not expect to find pupil-BOLD coupling. The same procedure as described above was carried out to move these masks from MNI space into our study-specific template space (see *Figure 2—figure supplement 2D* for individual examples).

## fMRI data quality assessment
### Comparisons with previous studies

We wanted to ensure that we could replicate the resting-state correlations between pupil size and BOLD patterns reported in previous studies. First, we followed the methods of *Schneider et al., 2016*. We convolved the pupil size time series with the canonical HRF (6 s TTP). This single-pupil regressor, along with the movement and physiological noise correction model, were entered into first-level GLMs. Note that we only refer to the light condition in *Schneider et al., 2016* since we did not assess other light conditions. Second, we followed the methods of *Murphy et al., 2014*. Here, we used the preprocessed pupil size time series and convolved that with the default canonical HRF, as well as its temporal and dispersion derivatives (*Friston et al., 2000*). The resulting three pupil time series were entered into a first-level GLM together with the movement and physiological noise correction model. For the *Murphy et al., 2014* comparison analysis, the first-level single-subject contrast maps were submitted to a second-level random-effects analysis (one-way ANOVA, three levels of pupil/basis functions). To interrogate pupil correlations within the LC, statistics were also carried at the second-level using small volume correction with our group LC mask and the LC mask (*Keren et al., 2009*) used by Murphy and colleagues as an ROI in SPM12. In line with the reports of *Schneider et al., 2016* and *Murphy et al., 2014*, the analyses described here included spatial smoothing with a 6 mm full-width at half maximum (FWHM) Gaussian kernel.

## Assessment of the quality of subcortical fMRI data

Next, we assessed the signal quality within the subcortical nuclei to ensure we would be able to capture pupil-AAS coupling. First, we inspected the tSNR of our data in all cortical and subcortical ROIs. To do this, the tSNR was calculated as the ratio of the mean and the standard deviation of the signal across the unsmoothed BOLD time series from the two sessions. We then averaged the resulting tSNR within each ROI. Second, we investigated whether we could replicate previous work reporting co-fluctuations between activity in various subcortical ROIs during rest (*van den Brink et al., 2019*). The extracted BOLD signal from each ROI (LC, VTA, SN, DR, MR, BF, and pons as a control region) per session, per participant, was denoised (using the movement and physiological noise correction model described above) and demeaned and then entered into a partial correlation analysis. We computed a partial correlation for each pair of AAS nuclei, controlling for activity in the pons. Correlation coefficients underwent a Fisher *r*-to-*Z* transform and were then submitted to one-sample *t*-tests.

## Pupil-AAS coupling analyses

To systematically examine pupil-AAS coupling and understand the temporal relationship between the two signals, we conducted a set of three main analyses. The rationale for our approach was that previous studies (see 'Comparisons with previous studies') assumed that pupil-brain coupling during rest would follow the canonical HRF used in event-related fMRI designs. However, these assumptions may not be correct or may not apply to subcortical nuclei. Therefore, in our first analysis we (i) convolved the pupil time series with participant-specific and ROI-specific estimates of the HRF (described in 'Estimation of region- and participant-specific HRF'), which showed a range of TTPs. In the second analysis, we (ii) systematically changed the TTP of the canonical HRF (from 1 to 6 s, in steps of 1 s) and convolved the pupil time series with each of these. Lastly, we (iii) performed a pupil-AAS cross-correlation analysis in which we did not convolve the pupil time series at all. Note that analyses (ii) and (iii) were not preregistered as they were carried out to better understand the outcome of analysis (i). Therefore, they should be deemed exploratory. We will now provide a detailed description of each of these analyses.

## Estimation of region- and participant-specific HRF

In the first main analysis, we aimed to account for HRF variability across different brain regions and participants in our resting-state data. Here, we deviated from the preregistration, in which we stated that we would obtain participant-specific HRFs from event-related fMRI data. However, the HRFs based on these event-related fMRI data did not provide plausible HRFs in the AAS regions (i.e., did not rise up to one tall peak and follow with an undershoot), possibly because these AAS regions may not have been involved in the task at hand. Instead, we used a blind deconvolution technique developed by *Wu et al., 2013* to estimate region- and participant-specific HRFs based on the data from both resting-state sessions. This point process method has been validated on simulated as well as empirical data (*Rangaprakash et al., 2018*; *Wu et al., 2021*). It assumes that a common HRF is shared across various spontaneous point process events (i.e., random neural events) in a given voxel or ROI. After physiological correction, the cleaned BOLD signal $y(t)$ at a given voxel or ROI is considered as the convolution of the voxel/ROI-specific HRF $h(t)$ and spontaneous neural events $x(t)$

$$y(t) = x(t) \otimes h(t) + c + \varepsilon(t)$$

where $c$ is a constant term indicating the baseline magnitude of the BOLD response, $(t)$ is noise, and $\otimes$ denotes convolution. Spontaneous point process events $n(t)$ were identified as BOLD fluctuations of relatively large amplitude (one or more standard deviations away from the mean; see *Figure 3A*). Before identifying these events, we removed movement and physiological noise with the same set of 33 regressors as described above (see 'MRI data preprocessing'). We then applied a high-pass filter (1/128 Hz cut-off) and AR(1) serial autocorrelation corrections. These events were modeled as a train of Dirac delta functions given by

$$\hat{n}(t) = \sum_{r=0}^{\infty} \delta(t - \tau)$$

where $\delta\left(\mathrm{t}-\tau\right)$ is the delta function. The ROI-specific HRF $h\left(t\right)$ was then fitted according to $n\left(t\right)$ using a canonical HRF and two derivatives (temporal derivative and dispersion derivative; *Friston et al., 2000*). Once $h\left(t\right)$ was calculated, we obtained an approximation $n\left(t\right)$ of the neural signal from the observed data using a Wiener filter. ROI-specific HRFs (*Figure 3B*) were estimated for all AAS nuclei (i.e., LC, VTA, SN, DR, MR, BF) and two validation regions (i.e., ACC, OCC) and one control region (i.e., pons). To maximize the number of spontaneous neural events, HRFs were estimated based on the concatenated BOLD signals from the two sessions (see *Figure 3C* for the number of detected pseudo-events per ROI). These HRFs were then convolved with the two pupil time series (i.e., pupil size [*Figure 3E*] and pupil derivative [*Figure 3F*]), forming the final pupil regressors that were entered into the GLMs for this analysis. GLMs were made up of a single pupil regressor-of-interest in addition to our physiological noise correction model described above. Analysis for pupil size included nine GLMs per participant, dedicated to six AAS nuclei (LC, VTA, SN, DR, MR, BF) and the three validation and control regions (ACC, OCC, pons). Similarly, nine models formed the analysis for the pupil derivative.

## Systematic adjustment of HRF time-to-peak

To explore other possible temporal relationships between pupil size and AAS BOLD patterns, we carried out a second analysis of pupil-AAS coupling where the TTP of the HRF was systematically shifted in time (*Pais-Roldán et al., 2020*). This was done using the canonical HRF (*Friston et al., 2000*), which by default has a TTP (delay of response relative to onset) of 6 s. To explore pupil-AAS coupling at shorter lags, we compared six HRFs with TTPs varying between 1, 2, 3, 4, 5, and 6 s. These HRFs were created using spm_hrf() in SPM, where parameter *p(1)* which refers to 'delay of response (relative to onset)' was adjusted from 6 (default) to 1, 2, 3, 4, and 5, respectively. Note that the 6 s TTP corresponds to the TTP used in the analysis corresponding to *Schneider et al., 2016*. These six HRFs were then convolved with the two pupil time series (pupil size, pupil derivative) for each participant, resulting in 12 pupil regressors. Each pupil size regressor was then added to the physiological noise correction model, making up six GLMs per participant, each focusing on one TTP (i.e., 1 s, 2 s, 3 s, 4 s, 5 s, 6 s). Similarly, six GLMs were created for the pupil derivative.

## Analyses using unconvolved pupil time series

To further characterize the nature of pupil-AAS coupling, we carried out two analyses using the *unconvolved* pupil time series. Firstly, we performed a cross-correlation analysis (*Pais-Roldán et al., 2020*) in which the preprocessed (downsampled, demeaned, unconvolved) pupil time series (size and derivative) were shifted forward and backward relative to the BOLD signal (denoised using the movement and physiological noise correction model and demeaned). Note that the BOLD signal from each ROI was first averaged and then entered into the cross-correlation analysis. Correlation coefficients for each time lag underwent a Fisher *r*-to-*Z* transform and were then submitted to one-sample *t*-tests. This analysis is similar to the TTP analysis but used unconvolved pupil time series and allowed us to investigate both positive and negative lags between the pupil and BOLD signals, which was not possible with the TTP method. Secondly, in order to determine which frequencies were driving the observed pupil-BOLD cross-correlations, we estimated for each ROI the cross-spectral power density (*Yellin et al., 2015*), the Fourier transform of the cross-correlations. We did this using cspd() in MATLAB, setting the window length to 10 samples with an overlap of three samples.

## Statistical analyses

All first-level GLMs described above were constructed in SPM12 with session (session 1, session 2) as a within-subject factor (n = 56). For participants in which only one session could be included (i.e., due to pupil quality exclusion criteria or technical issues in a scanning session), a GLM was constructed using only one session (n = 14). All analyses carried out in this study was based on n = 70, unless otherwise mentioned.

Second-level analyses were carried out by extracting *t*-values from single-subject contrast maps generated from the first-level analyses. These weights were then submitted to a second-level random effects analysis (one-sample *t*-test) in R using 'stats' package. To correct for multiple comparisons, alpha levels (set at 0.05) were adjusted by controlling the FDR (*Benjamini and Hochberg, 1995*; R 'stats' package p.adjust (method = 'fdr')).

For the comparison analyses with *Schneider et al., 2016* and *Murphy et al., 2014*, single-subject contrast maps obtained from first-level analyses were entered into second-level random effects analyses (one-sample *t*-test for *Schneider et al., 2016* and a one-way repeated-measures ANOVA with three levels for *Murphy et al., 2014* in SPM12). Here, we used a cluster-forming voxel-level threshold of p<0.001 (uncorrected). Alpha was set at 0.05 whole-brain family-wise error (FWE) corrected at the cluster level using Gaussian random field theory-based methods as implemented in SPM12 (*Friston et al., 1996*).

## Acknowledgements

Data collection was funded by the Templeton World Charity Foundation, Inc network 'Survival Circuits Influences on Human Nature' (TWCF number: 0366) to LdV, Elizabeth Phelps and Joseph LeDoux. This publication is part of a Vici project (with project number VI.C.181.032) that is financed by the Dutch Research Council (NWO) awarded to Sander Nieuwenhuis. We would also like to thank Birte Forstmann and Steven Miletić for helpful discussion.

## Additional information

### Funding

| Funder | Grant reference number | Author |
| --- | --- | --- |
| Templeton World Charity Foundation | 'Survival Circuits Influences on Human Nature' (TWCF number: 0366) | Lycia D de Voogd |
| Nederlandse Organisatie voor Wetenschappelijk Onderzoek | VI.C.181.032 | Sander Nieuwenhuis |

The funders had no role in study design, data collection and interpretation, or the decision to submit the work for publication.

### Author contributions

Beth Lloyd, Conceptualization, Data curation, Formal analysis, Validation, Investigation, Visualization, Methodology, Writing - original draft, Writing – review and editing; Lycia D de Voogd, Conceptualization, Resources, Data curation, Software, Supervision, Funding acquisition, Investigation, Project administration, Writing – review and editing; Verónica Mäki-Marttunen, Conceptualization, Supervision, Methodology, Writing – review and editing; Sander Nieuwenhuis, Conceptualization, Resources, Supervision, Funding acquisition, Methodology, Writing – review and editing

### Author ORCIDs

Beth Lloyd http://orcid.org/0000-0003-4034-8119
Sander Nieuwenhuis http://orcid.org/0000-0003-2418-3879

### Ethics

Human subjects: All human subjects gave informed consent and consent to publish. Human subjects participated in the experiment in accordance with the guidelines and regulations of the University Committee on Activities Involving Human Subjects at New York University (Institutional Review Board #2016-2).

### Decision letter and Author response

Decision letter https://doi.org/10.7554/eLife.84822.sa1
Author response https://doi.org/10.7554/eLife.84822.sa2

## Additional files

### Supplementary files
• MDAR checklist

## Data availability

The raw data are publicly available here: https://doi.org/10.5061/dryad.7m0cfxpzn. All numerical data and brain maps underlying all figures, tables, and figure supplements as well as the code to generate these figures and analyses are available here: https://osf.io/5rjcf/ (with a back-up on https://github.com/bethlloyd/Lloyd_etal_rsBOLD_pupil). Code to preprocess the fMRI and pupil data can be found here: https://github.com/bethlloyd/Lloyd_etal_rsBOLD_pupil (copy archived at *Lloyd, 2023*).

The following datasets were generated:

| Author(s) | Year | Dataset title | Dataset URL | Database and Identifier |
|---|---|---|---|---|
| Lloyd B, de Voogd LD, Mäki-Marttunen V, Nieuwenhuis S | 2023 | Pupil size reflects activation of subcortical ascending arousal system nuclei during rest | https://doi.org/10.5061/dryad.7m0cfxpzn | Dryad Digital Repository, 10.5061/dryad.7m0cfxpzn |
| Lloyd B, de Voogd LD, Maki-Marttunen V, Nieuwenhuis S | 2022 | The relationship between pupil size and locus coeruleus activity and other brainstem nuclei during rest | https://osf.io/5rjcf/ | Open Science Framework, 5rjcf |

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
