## [Editor Report]

These are important findings that show that pupil size is not only governed by the locus coeruleus but also by other neuromodulatory subcortical systems. Furthermore, the authors demonstrate that using a standard hemodynamic response kernel is not appropriate for capturing the activity of these systems, at least at rest. Thus, this paper presents compelling evidence against two prevalent working assumptions among researchers in the field.

---

## [Decision Letter]

**Decision letter after peer review:**

Thank you for submitting your article "Pupil size reflects activation of subcortical ascending arousal system nuclei during rest" for consideration by *eLife*. Your article has been reviewed by 3 peer reviewers, including Mara Mather as the Reviewing Editor and Reviewer #1, and the evaluation has been overseen by Christian Büchel as the Senior Editor. The following individual involved in review of your submission has agreed to reveal their identity: Heidi I.L. Jacobs (Reviewer #3).

Essential revisions:

The reviewers have each pointed out various points that should be clarified in the paper to further strengthen it.

*Reviewer #1 (Recommendations for the authors):*

p. 11: I found this statement confusing, "These exploratory analyses suggest that fluctuations in pupil diameter have a much closer temporal relationship with changes in AAS-BOLD activity than what is characteristic of event-related responses." – this sounds like the authors are comparing the temporal relationship of pupil size to BOLD during spontaneous fluctuations and during task-based events. But their data do not compare these two situations and so I'm guessing that instead do the authors mean to compare the shorter TTPs seen in their results to what is characteristic of event-related responses as they are typically defined in the field (i.e., with a longer TTP). The statement should be clarified.

p. 21: "Please note that we deviated from the preregistration and did not apply 'scrubbing' in addition to the 33 nuisance regressors." – it would help to clarify why this change was made.

Which FDR correction method was used? Does it allow for likely inter-correlated nature of the different outcomes?

*Reviewer #2 (Recommendations for the authors):*

The multiple panels in figures 3e,f should be titled in the figure rather than use described in the caption. Since they are arbitrary units and the y axis is not matched between 3e and 3f, the y axis for each plot should be scaled to the range of the data in each plot to enable a clearer view of the variability in each trace.

Similarly, in figure 4b and c (and S2), consider setting the y-axis limits from -2 to 2 and if necessary plotting LC in a physically larger (not differently-scaled) panel with y-axis from -2.5 to 2.5 or putting "//" around the violin plots for the LC that go beyond the -2 to 2 range. Seeing those extrema are less important than seeing the weight of the distribution, and this will enable readers to focus on the trend for each plot as TTP is varied.

*Reviewer #3 (Recommendations for the authors):*

My main concerns or questions relate to the robustness and replicability of these findings and interpretations. It seems that subjects were scanned twice, and at some point, in the results, a certain set of analyses was replicated in the second scan session (Figure S3, cross-correlations). Did I miss it or were the other analyses not replicated? If not, it would be valuable to demonstrate the robustness or reliability of the findings. What was the correlation of the BOLD signal across the two visits?

Other questions that I have:

2) Registration is critical for these tiny regions and there are no anatomical boundaries to verify their localization. How did the authors verify anatomical accuracy of registration – not functional correlation as mentioned in the manuscript? It would also help if the authors can provide individuals examples (or a video)?

3) Figure 4: Correlations between TTP and pupil size were positive for all regions. Yet, the violin plot shows a large variability in correlations for the LC with substantial negative correlations as well (even at TTP=1s). What do these negative correlations reflect, and why is the range of correlations so much larger for the LC compared to the SN? Could this reflect contributions of noise coming from the fourth ventricle CSF or could this be related to error due to the small functional masks (1-7 voxels according to the methods)?

4) Related to this: can the authors provide voxel count/volume of all ROIs examined?

5) As indicated, animal work has demonstrated an important role for the locus coeruleus in pupil dilation. An important question would be to determine whether the observed results are independent of the LC (i.e. which effects would remain present when regressing out LC BOLD signal). This could inform the field on the unique contribution of the LC or is shared contribution with other nuclei.

6) Lastly: Interestingly, the short Time to Peak was not uniquely observed for the neuromodulatory subcortical nuclei, but also in the cortex. How do the authors interpret this finding?

---

## [Author Response]

Essential revisions:The reviewers have each pointed out various points that should be clarified in the paper to further strengthen it.Reviewer #1 (Recommendations for the authors):p. 11: I found this statement confusing, "These exploratory analyses suggest that fluctuations in pupil diameter have a much closer temporal relationship with changes in AAS-BOLD activity than what is characteristic of event-related responses." – this sounds like the authors are comparing the temporal relationship of pupil size to BOLD during spontaneous fluctuations and during task-based events. But their data do not compare these two situations and so I'm guessing that instead do the authors mean to compare the shorter TTPs seen in their results to what is characteristic of event-related responses as they are typically defined in the field (i.e., with a longer TTP). The statement should be clarified.

We thank the reviewer for this comment and agree that the statement is confusing. We indeed mean that during rest, HRFs with short TTPs are associated with stronger correlations between pupil size and AAS BOLD activity compared to the broad HRFs typically used to model event-related responses (i.e., with longer TTPs). In line with the reviewer’s suggestion, we have clarified the statement by changing the sentence to the following:

For the adjusted sentence, see Results >> Positive pupil-AAS coupling using HRFs with rapid time-to-peaks [p 11, line 268-271].

“These exploratory analyses suggest that positive coupling between fluctuations in pupil diameter and AAS BOLD signal can be found when using a shorter transfer function (i.e., using TTPs of 1 to 2s), but not with the broad HRFs that are typically used to model event-related BOLD responses (i.e., with a longer TTP; Friston et al., 2000).”

p. 21: "Please note that we deviated from the preregistration and did not apply 'scrubbing' in addition to the 33 nuisance regressors." – it would help to clarify why this change was made.

We agree with the reviewer that we should clarify why we did not perform scrubbing. To clarify, we left this step out mainly for simplicity reasons. In our initial preregistered analysis (replication of Murphy et al., 2014) we performed the analysis with and without scrubbing as we stated in the preregistration (in the manuscript we only reported no scrubbing analysis for consistency). However, we did not observe a difference in the outcomes between the two approaches. This suggested that scrubbing, in our analysis pipeline, did not affect the outcomes as much as we expected. This may be because we already applied a rigorous cleaning method with RETROICOR (see also Kassinopoulos and Mitsis, 2022). For this reason, we opted to refrain from applying scrubbing in our subsequent exploratory analyses.

We added this clarification to the manuscript:

For the adjusted sentence, see Materials and methods >> MRI data preprocessing [p 21, line 562-564].

“An additional regressor was added to remove signal fluctuations from the fourth ventricle, which was manually delineated using individual MP-RAGE scans (M = 90, SD = 34 voxels). This movement and physiological noise correction model was added to all general linear models (GLMs) described below. Please note that since scrubbing did not affect the outcomes of our initial preregistered analysis (replication of Murphy et al., 2014), we report only the analysis without scrubbing and opted to refrain from applying scrubbing in the subsequent exploratory analyses.”

Which FDR correction method was used? Does it allow for likely inter-correlated nature of the different outcomes?

We would like to thank the reviewer for asking this question. The FDR correction method that we used was the Benjamini and Hochberg (1995) method. As we understand, this ‘BH’ FDR-correction procedure (R stats package p.adjust(method = ‘fdr’)) has been investigated with regards to test-statistics dependency in two studies: Benjamini and Yekutieli (2001) and Benjamini, Hochberg and Kling (1997). Both studies show that the FDR (in the BH procedure) controlled for positively correlated test statistics.

We thank the reviewer for pointing this out and we now added which FDR correction we used to the manuscript:

For the adjusted sentence, see Materials and methods >> Statistical analyses [p 27, line 756-757].

“To correct for multiple comparisons, α levels (set at 0.05) were adjusted by controlling the false discovery rate (FDR; Benjamini and Hochberg, 1995; R ‘stats’ package p.adjust(method = ‘fdr’)).”

Reviewer #2 (Recommendations for the authors):The multiple panels in figures 3e,f should be titled in the figure rather than use described in the caption. Since they are arbitrary units and the y axis is not matched between 3e and 3f, the y axis for each plot should be scaled to the range of the data in each plot to enable a clearer view of the variability in each trace.

We agree with the reviewer that Figure 3 (especially panels e and f) would be easier to read if the titles were more clearly placed and not only part of the axis description. We have taken on board the advice and placed titled plots within the figure, where panels e and f now clearly show which pupil index is where and which timeseries corresponds to which pupil preprocessing step. We also adjusted the y-axes in panels e and f, showing more clearly the variability for each pupil time-series.

Similarly, in figure 4b and c (and S2), consider setting the y-axis limits from -2 to 2 and if necessary plotting LC in a physically larger (not differently-scaled) panel with y-axis from -2.5 to 2.5 or putting "//" around the violin plots for the LC that go beyond the -2 to 2 range. Seeing those extrema are less important than seeing the weight of the distribution, and this will enable readers to focus on the trend for each plot as TTP is varied.

We appreciate the author bringing up this idea. We agree that the violin plots within Figure 4 were small and maybe difficult to inspect. We have taken on the reviewer’s idea to adjust the scaling of the LC and alter the y-axis for the remaining ROIs to the -2 to 2 range. Please note that the scaling between the LC and the other subcortical ROIs is in line, where -2 on the LC plot corresponds directly to -2 in the other subcortical ROI plots.

Please see Figure 4 in the manuscript for the updated plot [p 12].

Reviewer #3 (Recommendations for the authors):My main concerns or questions relate to the robustness and replicability of these findings and interpretations. It seems that subjects were scanned twice, and at some point, in the results, a certain set of analyses was replicated in the second scan session (Figure S3, cross-correlations). Did I miss it or were the other analyses not replicated? If not, it would be valuable to demonstrate the robustness or reliability of the findings. What was the correlation of the BOLD signal across the two visits?

We thank the reviewer for bringing up this point. As the cross-correlation is in essence similar to the time-to-peak analyses, we did not show the results of that analysis separately for session 1 and 2. However, to address the reviewer’s question, we now also examined the time-to-peak results in sessions 1 and 2 separately. As can be seen in Figure 4 —figure supplement 1, a similar general statistical pattern is present across both sessions, where pupil-AAS BOLD signal coupling is mainly present at early lags. Importantly, this analysis also shows that smaller ROIs, in particular the LC, DR, and MR, are more prone to a drop in statistical power when the data is halved (i.e., one session) compared to the original analysis (two sessions combined).

We have added these results to the supplementary materials (Figure 4 —figure supplement 1). In addition, we have added a part to the Results section (see below) to indicate that the time-to-peak analysis was also carried out on each session separately, and to summarise the results.

For the information about this analysis, see section: Results >> Positive pupil-AAS coupling using HRFs with rapid time-to-peaks [p 10, line 244-247].

“For the OCC ROI, we found a positive correlation at the 1-s TTP, followed by a shift to negative correlations at later TTPs (4 s to 6 s), which is in line with previous work (Breeden et al., 2016; Schneider et al., 2016; Yellin et al., 2015) and the results we reported above (*Whole-brain pupil-BOLD patterns consistent with previous studies),* whereas the ACC correlated positively with pupil size at predominantly early TTPs (1 s to 4 s; Figure 4C; 4D), similar to the AAS ROIs. To ensure the robustness of these findings, we repeated the analyses for each session separately. The pattern of results was visually and statistically similar between the two sessions, and in line with the results when both sessions were combined (Figure 4 —figure supplement 1).

Similar analyses for the pupil derivative also showed significant differences in the strength of pupil-BOLD coupling across the TTPs for the VTA (p = 0.012), SN (p = 0.022), DR (p = 0.002), ACC (p = 0.009), and OCC (p < 0.001; FDR-corrected for nine ROIs).”

Other questions that I have:2) Registration is critical for these tiny regions and there are no anatomical boundaries to verify their localization. How did the authors verify anatomical accuracy of registration – not functional correlation as mentioned in the manuscript? It would also help if the authors can provide individuals examples (or a video)?

We agree with the reviewer that accurate registration is crucial for extracting signal from these small subcortical nuclei. For that reason, we chose to use the ANTs SyN algorithm for all non-linear registrations (e.g., warping), since this was found to be among the best performing methods for normalization (Ewert et al., 2019; Klein et al., 2009).

To check that individual T1 images were indeed warped accurately to the template, we calculated the average and standard deviation of the individual warped T1 images (n = 72). The average (Figure 2 —figure supplement 2B) is slightly blurrier than the template image (Figure 2 —figure supplement 2A) because certain individual characteristics, particularly within cortical regions, remain. The registration accuracy is also clearly shown in the standard deviation image of the individual warped T1 images (Figure 2 —figure supplement 2C) whereby light areas (high values) indicate where the registration is more variable, while dark areas (low values) indicate low variability across the T1s. Overall, these images show that the brainstem/subcortical areas are well-registered (i.e., dark areas in the standard deviation image).

However, these images do not yet indicate accurate registration between the data and the ROIs. Therefore, to show the accuracy of the anatomical registration, we presented a handful of individual warped T1s with their ROI labels overlaid on top (see Figure 2 —figure supplement 2D). These examples also demonstrate the accuracy of the registration between the template and individual T1s. Please note that all ROIs, except the LC, were warped from MNI-space to template space using antsRegistrationSyN.sh. The accuracy of this registration was visually checked using anatomical boundaries as reference regions (see Author response image 1: T1_template_ -> T1_MNI-space_ registration check). The LC ROI relied on the transformations from individual T1 space to the template image described above, and shown in Figure 2 —figure supplement 2A-C.

**Author response image 1. sa2fig1:** T1_template_ -> T1_MNI-space_ registration check. Template image warped into MNI-space (yellow outline) overlaid on top of the original MNI-space T1.

Lastly, we visually verified anatomical accuracy of the ROIs on the T1s following published guidelines (Beliveau et al., 2015; Trutti et al., 2021) and by visually inspecting the original MNI-space ROIs and our transformed ROIs (Alkemade et al., 2020; Zaborszky et al., 2008).

For the information about this figure, see Materials and methods >> MRI data preprocessing [p 20, line 534-541].

“After the whole-brain template image *(T1_template_)* was generated, for optimal coregistration, all individual MP-RAGE scans (*T1_native space_*) were submitted to a new coregistration step (using antsRegistration.sh; Figure 2 —figure supplement 1B). For this we performed linear (rigid, then affine), followed by nonlinear (SyN), registration steps, resulting in optimized individual whole-brain scans in template space (*T1_group space_*). To ensure that individual MP-RAGE scans and the group template (T1_native space_ to T1_group space_) had been registered accurately, we carried out visual checks for each individual, using anatomical boundaries and ventricles as reference regions. We also calculated the average (Figure 2 —figure supplement 2B) and standard deviation (Figure 2 —figure supplement 2C) of the individual warped T1 images (n = 72). On the individual T1s standard deviation image (Figure 2 —figure supplement 2C), light areas (high values) indicate where the coregistration is more variable, while dark areas (low values) indicate low variability across the T1s. Overall, these images show that the brainstem/subcortical areas are well-registered (i.e., dark areas in the standard deviation image).

Mutual information maximization-based rigid-body registration was used to register MP-RAGE scans and functional images. Functional images were motion-corrected using rigid-body transformations.”

We also added the following information in Method >> Definition of regions-of-interest (ROIs) [p 23, line 624-626].

“All atlases were originally in MNI space. To move the atlases into our study-specific template space, antsRegistration.sh (using the same parameters as described above) was applied to generate the transformation matrices between MNI space and template space, which were applied to each ROI mask. Each subcortical mask was then thresholded and resliced to the functional space. We visually verified anatomical accuracy of the ROIs on the T1s following published guidelines (Beliveau et al., 2015; Trutti et al., 2021) and by visually inspecting the original MNI-space ROIs and our transformed ROIs (Alkemade et al., 2020; Zaborszky et al., 2008).”

We refer to Figure 2 —figure supplement 2D in Methods >> Definition of regions-of-interest (ROIs) [p 23, line 637-638].

**“**The same procedure as described above was carried out to move these masks from MNI space into our study-specific template space (see Figure 2 —figure supplement 2D for individual examples).”

3) Figure 4: Correlations between TTP and pupil size were positive for all regions. Yet, the violin plot shows a large variability in correlations for the LC with substantial negative correlations as well (even at TTP=1s). What do these negative correlations reflect, and why is the range of correlations so much larger for the LC compared to the SN? Could this reflect contributions of noise coming from the fourth ventricle CSF or could this be related to error due to the small functional masks (1-7 voxels according to the methods)?

We have also noticed that certain ROIs, in particular the LC, show larger inter-individual variability in pupil-BOLD signal correlations than other ROIs, such as the SN. Indeed, it is possible that the variability is due to the size of the mask. However, we did not observe a significant correlation between a participant’s *t*-statistic (i.e., for the LC at TTP = 1s) and the size of their LC mask (Spearman’s rho(68) = 0.11, *p* = 0.38). Nor was there a significant correlation between the *t*-statistic and the tSNR in the LC (session 1: Spearman’s rho(68) = 0.12, *p* = 0.34; session 2: Spearman’s rho(68) = -0.06, *p* = 0.63). Lastly, we checked whether participants showing a negative correlation in one session also showed a negative correlation in the other session, but this was not the case (Spearman’s rho(54) = 0.18, *p* = 0.19; *n* = 56 [only participants with both sessions included]). We also recognize the possibility that the negative pupil-BOLD correlations could reflect noise from surrounding ventricles. However, to minimize this concern, we employed rigorous fMRI cleaning methods that accounted for signal variance from the 4th ventricle.

In sum, we cannot explain the inter-individual variability in pupil-BOLD signal correlations by differences in mask size or tSNR. Therefore, it remains an open question what drives these individual differences in pupil-LC BOLD signal correlations.

We added these additional analyses to the Results >> Positive pupil-AAS coupling using HRFs with rapid time-to-peaks [p 11, line 259-267].

“The same analyses carried out for the control region in the pons revealed no main effect of TTP for pupil size or the pupil derivative, nor were there any positive or negative associations with pupil size or the pupil derivative for any TTP, attesting to the specificity of the pupil-BOLD associations found in our AAS and cortical ROIs. Statistical parametric maps including whole brain results for all TTPs are shown in Figure 4C and Figure 4 —figure supplement 2B.

Observing Figure 4, it seems that the LC exhibited greater inter-individual variability in pupil-BOLD signal correlations than most other ROIs. To explore the reasons for these differences, we investigated the correlation between a participant’s *t*-statistic (i.e., for the LC at TTP = 1s) and the size of their LC mask, as well as the tSNR in the LC. However, we did not find any significant correlation between the *t*-statistic and mask size (Spearman's rho(68) = 0.11, *p* = 0.38) or tSNR (session 1: Spearman's rho(68) = 0.12, *p* = 0.34; session 2: Spearman's rho(68) = -0.06, *p* = 0.63). Additionally, we examined whether participants with a negative correlation in one session also exhibited a negative correlation in the other session, but this was not the case (Spearman’s rho(54) = 0.18, *p* = 0.19; n = 56 [participants with both sessions included]). Therefore, it remains an open question what drives these individual differences in pupil-LC BOLD signal correlations.

These exploratory analyses suggest that positive coupling between fluctuations in pupil diameter and AAS BOLD signal can be found when using a shorter transfer function (i.e., using TTPs of 1 to 2s), but not with the broad HRFs that are typically used to model event-related BOLD responses (i.e., with a longer TTP; Friston et al., 2000).”

4) Related to this: can the authors provide voxel count/volume of all ROIs examined?

We thank the reviewer for pointing out this information was missing. We have now provided the voxel count for all ROIs examined in functional space.

For this additional information, see section: Materials and methods >> Definition of regions-of-interest (ROIs), [p 23, line 616-619].

“Published probabilistic atlases were used for the remaining subcortical ROIs as there are no established protocols for individual segmentation of these regions: VTA (Trutti et al., 2021; size in functional space = 45 voxels), SN (Alkemade et al., 2020; 84 voxels), DR (Beliveau et al., 2015; 4 voxels), MR (Beliveau et al., 2015; 3 voxels) and the sublenticular (Ch4) part of the BF (Eickhoff et al., 2005; Zaborszky et al., 2008; 31 voxels), …”

and [p 23, line 632-635].

“Therefore, we included these two cortical regions as additional validation ROIs. Specifically, we obtained masks of the calcarine sulcus in the OCC (size in functional space = 1975 voxels) and ACC (1148 voxels) using the automated anatomical labeling atlas in SPM (Tzourio-Mazoyer et al., 2002). Lastly, to explore the specificity of our pupil-AAS BOLD results, we delineated a cubic ROI in the medial part of the basis pontis (pons; 8 voxels), which served as a control region in which we did not expect to find pupil-BOLD coupling.”

5) As indicated, animal work has demonstrated an important role for the locus coeruleus in pupil dilation. An important question would be to determine whether the observed results are independent of the LC (i.e. which effects would remain present when regressing out LC BOLD signal). This could inform the field on the unique contribution of the LC or is shared contribution with other nuclei.

We agree with the reviewer that this is an important question. In accordance with the reviewer’s suggestion, we computed partial correlations between pupil size and BOLD signal while partialling out LC BOLD signal using linear regression. We focused on time lag = 0s (cross-correlation analysis), where we find overall strongest pupil-BOLD coupling.

Interestingly, as shown in Figure 5 in the manuscript, partialling out the contribution of the LC resulted in a slight but significant drop for the BF (*p* = 0.028). The signal from the VTA, SN, MR and BF remained significantly correlated with pupil size (*p*s <.030).

The information about this analysis has been added to the Results >> Positive pupil-AAS coupling when BOLD patterns closely follow pupil fluctuations [p 13, line 310-315].

“Similarly, we found that both ACC and OCC correlated most strongly with the pupil derivative (Figure supplement 4) at relatively short positive lags (0 s to +4 s), with a shift to a strong negative correlation at maximum positive lags (+8 s), especially in the OCC.

To determine whether the observed pupil-AAS BOLD signal cross-correlation results were independent of the LC, we recomputed the correlations between pupil size and BOLD signal after partialling out the contribution of the LC BOLD signal using linear regression. As shown in Figure 5C, partialling out the contribution of the LC resulted in a slight but significant drop for the BF (*p* = 0.028). The signal from the VTA, SN, MR and BF remained significantly correlated with pupil size (*p*s <.030). These results contribute further evidence to the notion that pupil size is not selectively driven by the LC, but that a broader network of AAS nuclei is involved.

Together, the TTP analysis and cross-correlation analysis yield essentially the same outcome, suggesting that no HRF convolution is needed to characterize the relationship between pupil size and AAS BOLD patterns during rest.”

We also added a sentence in the Discussion referring to these findings [p 16, line 367-369]:

“We found robust positive correlations between pupil size and BOLD signal in five of our AAS ROIs: LC, VTA, SN, MR and (sublenticular) BF. The correlations with the VTA, SN, MR and BF BOLD signals remained significant after controlling for the effect of the LC BOLD signal, suggesting that these AAS nuclei have independent influences on pupil size. The positive relationship with pupil size was less robust for the DR, and only significant in the TTP analyses, perhaps because this area had lower tSNR than the other subcortical ROIs.”

Additionally, as a sanity check, we repeated these partial correlation analyses but now partialling out the contribution of BOLD signal from the VTA instead of the LC. Since the VTA and SN show strong coupling (shown in Figure 2C), we would expect, in particular, that the mean pupil size-SN BOLD signal correlation would decrease after regressing out VTA BOLD activity. Indeed (see Author response image 2), pupil-SN coupling dropped significantly from full to partial correlations (two-tailed paired *t*-test: *p* <.001), although the partial correlation was still significantly above zero (*p* <.001).

6) Lastly: Interestingly, the short Time to Peak was not uniquely observed for the neuromodulatory subcortical nuclei, but also in the cortex. How do the authors interpret this finding?

We agree with the reviewer that this is an interesting finding. Only a handful of studies have examined the correlations between pupil size and cortical BOLD signal during rest (Breeden et al., 2016; DiNuzzo et al., 2019; Murphy et al., 2014; Schneider et al., 2016; Yellin et al., 2015). Of these, only one (Yellin et al., 2015) attempted to interrogate the temporal relationship between pupil size, BOLD signal and (implicitly) the underlying neuronal activity. Yellin and colleagues explored pupil-BOLD correlations at different time lags, similar to our approach, in a number of cortical ROIs (Figure 3c in Yellin et al. [2015]). Similar to us, they found that BOLD signal in their default mode network ROIs, in particular the inferior parietal lobule, correlated most strongly with pupil size at lags of ~2s; and that pupil size and OCC BOLD signal were negatively coupled at lags of ~5s.

We agree with the reviewer that it is important to recognise the close coupling between pupil size and our cortical ROIs, so we have added the following section to the manuscript:

Section: Results >> Positive pupil-AAS coupling using HRFs with rapid time-to-peaks [p 10, line 239-244].

“Specifically, we found positive correlations for all AAS regions at earlier TTPs (especially the 1-s [Figure 4B] and 2-s TTPs) but no significant correlations (LC, VTA, SN, DR, MR) at later TTPs (5 s to 6 s; Figure 4A). For the OCC ROI, we found a positive correlation at the 1-s TTP, followed by a shift to negative correlations at later TTPs (4 s to 6 s), which is in line with previous work (Breeden et al., 2016; Schneider et al., 2016; Yellin et al., 2015) and the results we reported above (*Whole-brain pupil-BOLD patterns consistent with previous studies*). The ACC correlated positively with pupil size at predominantly early TTPs (1 s to 4 s; Figure 4CD), similar to the AAS ROIs, and in line with pupil-BOLD coupling in default-mode network areas (Yellin et al., 2015). To ensure the robustness of these findings, we repeated the analyses for each session separately. The pattern of results was visually and statistically similar between the two sessions, and in line with the results when both sessions were combined (Figure 4 —figure supplement 1).”

References

Alkemade, A., Mulder, M. J., Groot, J. M., Isaacs, B. R., van Berendonk, N., Lute, N., Isherwood, S. J., Bazin, P.-L., and Forstmann, B. U. (2020). The Amsterdam Ultra-high field adult lifespan database (AHEAD): A freely available multimodal 7 Tesla submillimeter magnetic resonance imaging database. *NeuroImage*, *221*, 117200. https://doi.org/10.1016/j.neuroimage.2020.117200

Beliveau, V., Svarer, C., Frokjaer, V. G., Knudsen, G. M., Greve, D. N., and Fisher, P. M. (2015). Functional connectivity of the dorsal and median raphe nuclei at rest. *NeuroImage*. https://doi.org/10.1016/j.neuroimage.2015.04.065

Benjamini, Y., and Hochberg, Y. (1995). Controlling the False Discovery Rate: A Practical and Powerful Approach to Multiple Testing Author ( s ): Yoav Benjamini and Yosef Hochberg Source: Journal of the Royal Statistical Society. Series B (Methodological), Vol. 57 , No. 1 (995), Publi. *Journal of the Royal Statistical Society*, *57*(1), 289–300.

Benjamini Y. and Yekutieli D. (2001). The control of the false discovery rate in multiple testing under dependency. *The Annals of Statistics*, *29*(4), 1165–1188.

Benjamini Y., Hochberg, Y. and Kling, Y. (1997). False discovery rate control in multiple hypotheses testing using dependent test statistics. Research Paper 97-1, Dept. Statistics and O.R., Tel Aviv Univ.

Breeden, A. L., Siegle, G. J., Norr, M. E., Gordon, E. M., and Vaidya, C. J. (2016). Coupling between spontaneous pupillary fluctuations and brain activity relates to inattentiveness. *The European Journal of Neuroscience*. 1–7. https://doi.org/10.1111/ejn.13424

DiNuzzo, M., Mascali, D., Moraschi, M., Bussu, G., Maugeri, L., Mangini, F., Fratini, M., and Giove, F. (2019). Brain Networks Underlying Eye’s Pupil Dynamics. *Frontiers in Neuroscience*. https://doi.org/10.3389/fnins.2019.00965

Ewert, S., Horn, A., Finkel, F., Li, N., Kühn, A. A., and Herrington, T. M. (2019). Optimization and comparative evaluation of nonlinear deformation algorithms for atlas-based segmentation of DBS target nuclei. *NeuroImage*. https://doi.org/10.1016/j.neuroimage.2018.09.061

Friston, K. J., Mechelli, A., Turner, R., and Price, C. J. (2000). Nonlinear responses in fMRI: The balloon model, Volterra kernels, and other hemodynamics. *NeuroImage*. https://doi.org/10.1006/nimg.2000.0630

Kassinopoulos, M., and Mitsis, G. D. (2022). A multi-measure approach for assessing the performance of fMRI preprocessing strategies in resting-state functional connectivity. *Magnetic Resonance Imaging*, *85*(October 2021), 228–250. https://doi.org/10.1016/j.mri.2021.10.028

Klein, A., Andersson, J., Ardekani, B. A., Ashburner, J., Avants, B., Chiang, M. C., Christensen, G. E., Collins, D. L., Gee, J., Hellier, P., Song, J. H., Jenkinson, M., Lepage, C., Rueckert, D., Thompson, P., Vercauteren, T., Woods, R. P., Mann, J. J., and Parsey, R. V. (2009). Evaluation of 14 nonlinear deformation algorithms applied to human brain MRI registration. *NeuroImage*, *46*(3), 786–802. https://doi.org/10.1016/j.neuroimage.2008.12.037

Murphy, P. R., O’Connell, R. G., O’Sullivan, M., Robertson, I. H., and Balsters, J. H. (2014). Pupil diameter covaries with BOLD activity in human locus coeruleus. *Human Brain Mapping*. https://doi.org/10.1002/hbm.22466

Schneider, M., Hathway, P., Leuchs, L., Sämann, P. G., Czisch, M., and Spoormaker, V. I. (2016). Spontaneous pupil dilations during the resting state are associated with activation of the salience network. *NeuroImage*. https://doi.org/10.1016/j.neuroimage.2016.06.011

Trutti, A. C., Fontanesi, L., Mulder, M. J., Bazin, P. L., Hommel, B., and Forstmann, B. U. (2021). A probabilistic atlas of the human ventral tegmental area (VTA) based on 7 Tesla MRI data. *Brain Structure and Function*. https://doi.org/10.1007/s00429-021-02231-w

Yellin, D., Berkovich-ohana, A., and Malach, R. (2015). NeuroImage Coupling between pupil fluctuations and resting-state fMRI uncovers a slow build-up of antagonistic responses in the human cortex. *NeuroImage*, *106*, 414–427. https://doi.org/10.1016/j.neuroimage.2014.11.034

Zaborszky, L., Hoemke, L., Mohlberg, H., Schleicher, A., Amunts, K., and Zilles, K. (2008). Stereotaxic probabilistic maps of the magnocellular cell groups in human basal forebrain. *NeuroImage*, *42*(3), 1127–1141. https://doi.org/10.1016/j.neuroimage.2008.05.055